# Error-related signaling in nucleus accumbens D2 receptor-expressing neurons guides inhibition-based choice behavior in mice

Tadaaki Nishioka [1,2] ✉, Suthinee Attachaipanich[1], Kosuke Hamaguchi [3], Michael Lazarus [4], Alban de Kerchove d'Exaerde[5], Tom Macpherson [1,6] ✉ & Takatoshi Hikida [1,6] ✉

Learned associations between environmental cues and the outcomes they predict (cue-outcome associations) play a major role in behavioral control, guiding not only which responses we should perform, but also which we should inhibit, in order to achieve a specific goal. The encoding of such cue-outcome associations, as well as the performance of cue-guided choice behavior, is thought to involve dopamine D1 and D2 receptor-expressing medium spiny neurons (D1-/D2-MSNs) of the nucleus accumbens (NAc). Here, using a visual discrimination task in male mice, we assessed the role of NAc D1-/D2-MSNs in cue-guided inhibition of inappropriate responding. Cell-type specific neuronal silencing and in-vivo imaging revealed NAc D2-MSNs to contribute to inhibiting behavioral responses, with activation of NAc D2-MSNs following response errors playing an important role in optimizing future choice behavior. Our findings indicate that error-signaling by NAc D2-MSNs contributes to the ability to use environmental cues to inhibit inappropriate behavior.

Choice behavior can be guided not only by strategies based upon the acquisition of a desirable outcome, but also by those aimed at reducing undesirable outcomes. Indeed, we are often faced with situations in which the correct behavioral response to acquire desired outcomes is unknown or ambiguous, but prior experience of failure can be used to inhibit inappropriate responses. The important role that negative outcomes play in guiding the inhibition of behavior has long been appreciated, dating back from Thorndike's (1927) law of effect to the more recent proposal of *loss aversion* in the field of behavioral economics[1–3]. However, the neural mechanisms that underlie the use of such inhibition-based strategies for choice behavior are still unclear, and studies to date have primarily focused on investigating the reinforcement of behaviors that directly result in rewarding outcomes[4–6].

Within a limbic cortico-basal ganglia-thalamo-cortical signaling loop, the nucleus accumbens (NAc), in particular the dorsolateral NAc subregion, is suggested to play an important role in decision-making via its role in linking information concerning outcome values with that related to goal selection[7–10]. NAc medium spiny projection neurons (MSN), the major neuron type, are typically anatomically divided into two roughly equal subpopulations; dopamine D1 receptor-expressing MSNs (D1-MSN) that project predominantly to the ventral pallidum (VP) and substantia nigra pars reticulata (SNr), and dopamine D2 receptor-expressing MSNs (D2-MSN) that project predominantly to the VP[11,12]. While previous studies have demonstrated NAc D1-MSNs to be implicated in reward-related learning and D2-MSNs to be involved in aversion-related learning and behavioral flexibility[13–15], the specific

[1]Laboratory for Advanced Brain Functions, Institute for Protein Research, Osaka University, Suita, Japan. [2]Laboratory for Developing Minds, Icahn School of Medicine at Mount Sinai, New York, NY, USA. [3]Department of Biological Sciences, Graduate School of Medicine, Kyoto University, Kyoto, Japan. [4]International Institute for Integrative Sleep Medicine (WPI-IIIS) and Faculty of Medicine, University of Tsukuba, Tsukuba, Japan. [5]Neurophysiology Lab, ULB Neuroscience Institute, Université Libre de Bruxelles, Brussels, Belgium. [6]These authors contributed equally: Tom Macpherson, Takatoshi Hikida. ✉e-mail: nishioka.tadaaki.88n@kyoto-u.jp; macpherson@protein.osaka-u.ac.jp; hikida@protein.osaka-u.ac.jp

role that NAc D1- and D2-MSNs may play in the inhibition of behavioral responses is less clear.

To isolate and measure the ability for inhibition-based choice behavior, we designed a visual discrimination-based cue-guided inhibition learning (VD-Inhibit) task in which mice were required to inhibit a touch response at a visual cue known to be associated with a reward omission, and instead respond at a random cue that had not previously been associated with any outcome in order to acquire a liquid reward. In a series of experiments, miniature microscope in vivo calcium imaging was used to investigate the precise activity patterns of D1- and D2-MSNs during the VD-Inhibit task, while time-specific optogenetic silencing of NAc D1-MSN or D2-MSNs during the same task were used to establish whether inactivation of these two subpopulations impairs the utilization of inhibition-based behavioral strategies. Our findings indicate that while D1-MSNs are bidirectionally modulated by rewarding outcomes and reward omission, D2-MSNs are activated by reward omission and are necessary for the inhibition of responses resulting in non-rewarded outcomes.

## Results

### Mice can acquire inhibition-based choice behavior

To assess the ability for inhibition-based choice behavior in mice, we modified a touchscreen-based visual discrimination task[4] and created an inhibition-based visual discrimination task (VD-Inhibit) in which mice used visual cues to determine which of two touchscreen response windows should be inhibited in order to receive a liquid reward by responding at the alternate window (Fig. 1a). Following trial initiation, a visual cue was presented in each of the two response windows (Fig. 1b); one visual cue (correct cue) was randomly changed each trial (51 possible images) and resulted in the delivery of a liquid reward (7 μl condensed milk at the reward magazine) following a touchscreen response, while the other visual cue (incorrect cue) was kept consistent during all trials and resulted in no reward and a 5-s time-out following a touchscreen response. This design meant that mice could not form a cue-outcome association for the correct cue due to its random nature, but must instead rely on the cue-outcome association of the incorrect cue to guide appropriate behavior (inhibition of the known visual cue and a touch response at the unknown visual cue).

Within 14 days of training, all C57BL/6J mice were able to reach the task criterion of ≥80% correct responses in a 60-min session for two consecutive days, indicating that visual cues signaling incorrect responses are sufficient to guide inhibition-based choice behavior (Fig. 1c, e, f). Indeed, as training progressed, the correct response latency gradually decreased, suggesting that mice were able to perform the task more efficiently following repeated training (Fig. 1d, g). On the other hand, there was no significant change in reward collection latency, suggesting that motivation to acquire the reward does not change across learning (Fig. 1d, h).

To compare the learning of this inhibition-based choice behavior with that of attendance-based choice behavior, a more classically used reward learning paradigm, we also created an attendance-based visual discrimination task in which a consistent visual cue signaled the rewarded response window that should be attended to, while a randomly-assigned visual cue signaled an unrewarded response window (VD-Attend task) (Fig. S1a and S1b). The number of sessions and total errors to criterion in the VD-Inhibit task were greater than that in the VD-Attend task (Fig. S1c, S1d, and S1e), suggesting that acquisition of an inhibition-based choice strategy is more difficult than that of an attendance-based choice strategy. However, the performances measured by the percentage of correct choices were comparable in the VD-Attend and VD-Inhibit tasks in the first and last sessions, indicating that following training, mice were able to effectively perform both tasks (Fig. S1f and S1g). The latencies to make a correct response and to collect the reward were also comparable between the VD-Attend and VD-Inhibit tasks (Fig. S1h and S1i).

Together, these data indicate the ability of mice to acquire response inhibition towards a visual cue signaling an incorrect response window, providing a framework for studying the neural mechanisms underlying inhibition-based choice behavior.

### NAc D1- and D2-MSN activity is especially modulated during the outcome period of correct and error trials

Given the proposed importance of the NAc in choice behavior and inhibitory control[10,16–18], we next investigated whether NAc D1- and D2-MSNs show changes in neural activity during different temporal windows of the VD-Inhibit task. To measure neural activity, we performed in vivo calcium imaging of NAc D1- and D2-MSNs at the single-cell level using a miniature microscope. An AAV expressing the fluorescent calcium indicator, jGCaMP7f[19], in a Cre-dependent manner (AAV9-hSyn-FLEX-jGCaMP7f) was microinjected into the dorsolateral NAc, a subregion proposed to be important for choice behavior[8,10,20,21] of D1-Cre or D2-Cre mice, and a gradient-index (GRIN) lens was implanted above the viral injection site (Figs. 2a, b, S2, and S3). Neural activity was recorded in freely moving mice on the second and criterion sessions of the VD-Inhibit task, and both D1- and D2-Cre mice were able to acquire the task within 1–3 weeks of training (Fig. S3). A constrained non-negative matrix factorization method for microendoscopic images (CNMFe[22]) was used to analyze the neural activity of individual NAc D1- or D2-MSNs during performance of the VD-Inhibit task (Fig. 2c, d; Supplementary Movie 1 and Supplementary Movie 2). In the criterion session, a total of 259 cells were identified in D1-Cre mice (62, 109, and 88 cells were identified from the three mice, respectively) and 194 cells in D2-Cre mice (61, 43, and 90 cells were identified from the three mice, respectively). To determine whether neurons' activities were modulated by correct or error responses, we performed hierarchical clustering and classified neurons into groups based upon their activity profiles in correct and error trials (Figs. 2e, f, S4, and S5). The averaged activities of identified neurons during the ITI (−5–0 s from trial initiation), Cue (Cue onset, 0–1.5 s from trial initiation; and Cue offset, −1.5–0 s from a response), and Outcome period (0-5 sec from a response) were then compared between correct and error trials to identify the time window during which activity was altered for each neuron type (Fig. 2g–k).

In NAc D1-MSNs, five groups were identified; Type I (17.0%, 44 of 259 cells), Type II (27.4%, 71 of 259 cells), Type III (8.5%, 22 of 259 cells), Type IV (29.3%, 76 of 259 cells), and Type V (17.8%, 46 of 259 cells) (Fig. 2f–k). Most notably, all D1-MSNs demonstrated a significant difference in activity between correct and error trials during the Outcome period, with Types I and IV in particular showing a marked difference between the two trial types characterized by a large increase or decrease, respectively, in activity during correct trials (Fig. 2g, j). Also of note, several D1-MSN types showed significant differences in activity between correct and error trials during ITI (Types I and IV) and Cue (Types I and IV) periods (Fig. 2g–k). Given that studies have shown that the ventral striatum encodes the average or net expected reward (or 'state value') during ITI periods[23,24], we next analyzed whether these ITI activities were modulated by past history. It was found that neural activity during the ITI in D1-Type I and Type IV neurons (Fig. 2g, j) was modulated by the results of previous trials (Fig. S6), with error responses in the previous trial associated with an increase or decrease, respectively, in ITI activity in current trials that resulted in response errors.

In D2-MSNs, a total of four groups were identified, Type I (12.9%, 25 of 194 cells), Type II (54.1%, 105 of 194 cells), Type III (6.7%, 13 of 194 cells), and Type IV (26.3%, 51 of 194 cells) (Fig. 3a, b). As with D1-MSNs, all D2-MSN types showed significantly different activity between correct and error trials during the Outcome period, with Types I and IV (Fig. 3c, f), in particular, demonstrating a large increase in activity during correct or error trials, respectively. Interestingly, these two populations also showed significantly different activity between

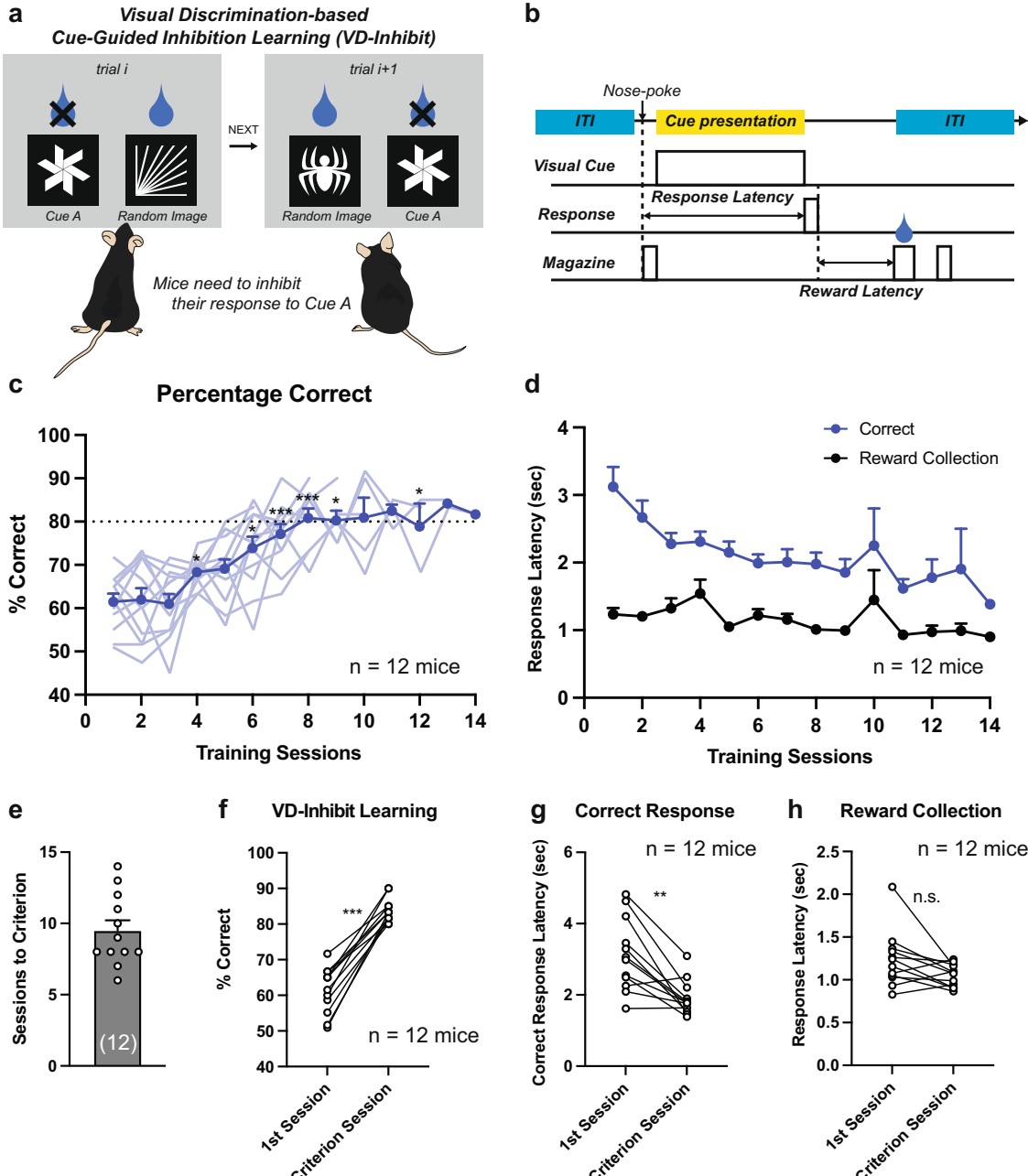

**Fig. 1 | Experimental design and behavioral performance. a** Experimental design. **b** Timeline of the task events and the definition of the behavioral parameters. **c** The percentage of correct in each session ($n = 12$ mice, One-way RM-ANOVA with Sidak correction, *$p < 0.05$ vs Session 1; **$p < 0.01$ vs Session 1, ***$p < 0.001$ vs Session 1). **d** Correct response latency and reward collection latency in each session. **e** Sessions to criterion. **f** The percentage of correct in the 1st and the last session (two-sided paired $t$-test, $t_{11} = 9.799$, ***$p < 0.0001$). **g** Correct response latency in the 1st and the last session (two-sided Wilcoxon signed rank test, **$p = 0.0024$). **h** Reward collection latency in the 1st and the last session (two-sided paired $t$-test, $t_{11} = 2.077$, $p = 0.0620$). Data are presented as mean ± SEM. The numbers of mice are shown in parentheses.

correct and error trials during the ITI period, which appeared to be the result of a slight decrease in activity during error trials. Finally, Types II showed a significant difference between correct and error trials during the Cue period of the trial (Fig. 3d). Similar to D1-MSNs, we also analyzed the effect of past history on neural activity during the ITI in D2-MSNs and found that neural activity in D2-Type I and Type IV neurons was modulated by the result (correct or error) of the previous trial (Fig. S7). Indeed, as with D1-MSNs, error responses in the previous trial were associated with a decrease or increase in activity in D2-Type I and Type IV neurons, respectively, during the ITI of current trials that resulted in response errors.

These data indicate that both D1- and D2-MSNs in the dorsolateral NAc signal the trial's outcome. Additionally, neuronal activity during the ITI was also modulated by response errors in the previous trial in specific D1- and D2-MSN types (Type I and Type IV), although it is still unclear as to how altered activity during the ITI period of some trials following a previous error trial predisposes animals to make a response error. Finally, the majority (~80%) of D2-MSNs were inhibited during the Outcome period of correct responses (Type II) or activated during the Outcome period of error responses (Type IV), with stronger neural activity on error than correct trials.

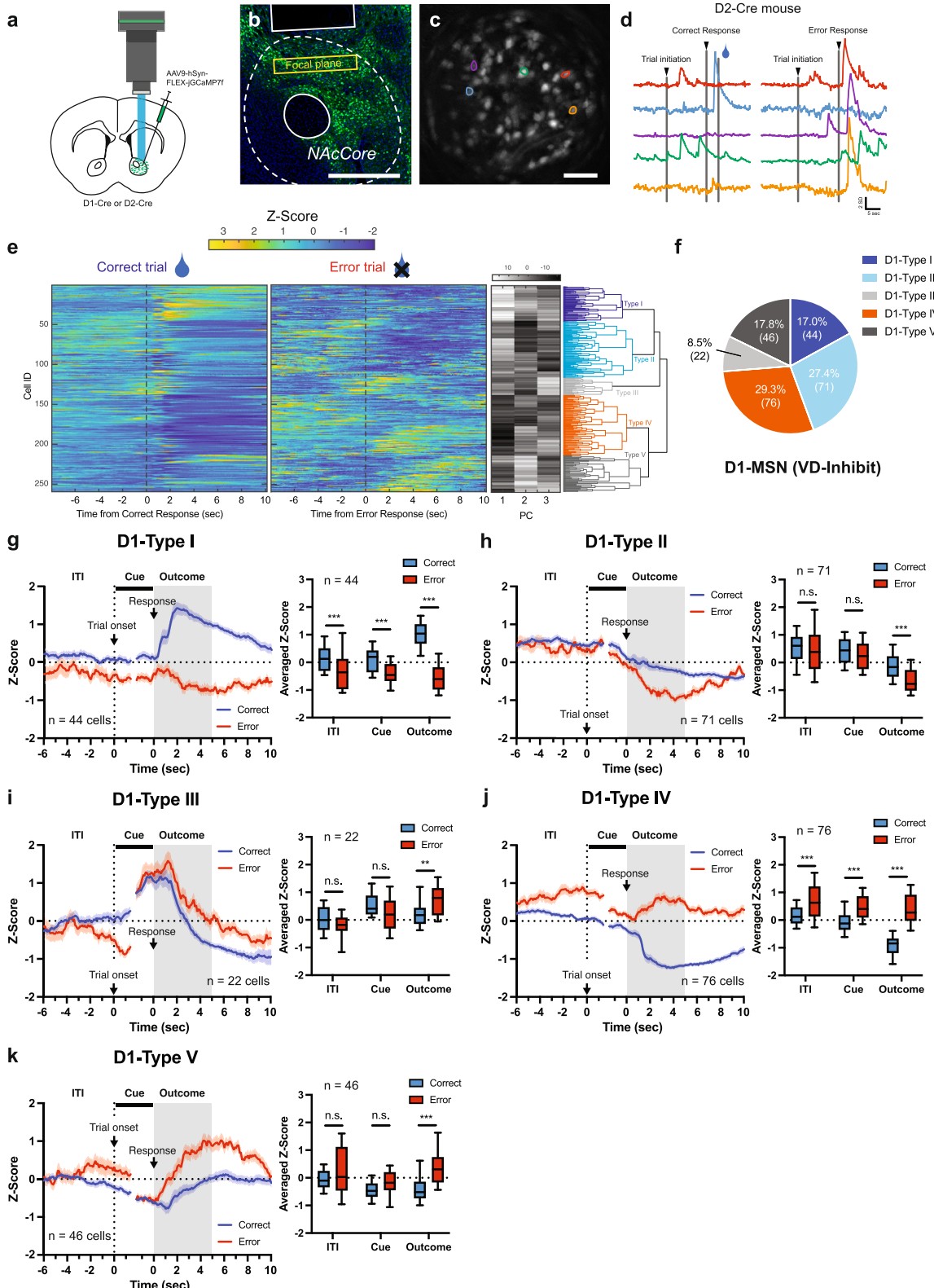

## NAc D2-MSNs generally conveys error information while D1-MSNs change the information they encode depending on the task

Next, we tested whether neural activity in D1- and D2-MSNs changes in a task-dependent manner. As with calcium imaging from mice performing the VD-Inhibit, we recorded from freely moving mice performing the VD-Attend task on the criterion sessions (Fig. S8a). To determine which neurons' activities were modulated by correct or error responses, we again performed hierarchical clustering and classified neurons into groups based upon their activity profiles.

Five groups of D1-MSNs were identified (Fig. S8b); Type I (19.8%, 25 of 126 cells), Type II (27.0%, 34 of 126 cells), Type III (21.4%, 27 of 126 cells), Type IV (20.7%, 26 of 126 cells), and Type V (11.1%, 14 of 126 cells). In the VD-Attend task too, all D1-MSN types demonstrated a significant

**Fig. 2 | Individual D1-MSNs are modulated by both response corrects and errors. a** A schematic of viral injection and GRIN lens implantation. **b** Representative coronal image of jGCaMP7f expression in D2-MSNs. Scale bar, 500 μm. **c** Maximum projection image of a representative imaging plane. Scale bar, 50 μm. **d** Example traces of individual neurons from a representative mouse performing the VD-Inhibit task. **e** Heatmap of neuronal activity of all D1 neurons recorded in the correct and error trials (Left). Each row represents trial-averaged calcium traces from one neuron. First three principal components (PC) and hierarchical clustering dendrogram showing the relationship of each neuron within the five clusters (Right). **f** Proportion of each type of cell in D1-MSNs. **g**–**k** Averaged traces of D1-MSN cell types in correct and error trials (Left) and averaged

Z-score during ITI (−5–0 s from trial onset), Cue (0–1.5 s from trial onset and −1.5–0 s from a response), and Outcome (0–5 s from response) period (Right, Two-way RM ANOVA with Sidak correction) for Type I (**g**, ITI, ***$p = 0.0004$; Cue, ***$p < 0.0001$; Outcome, ***$p < 0.0001$, $n = 44$ cells), Type II (**h**, ITI, $p = 0.6807$; Cue, $p = 0.3840$; Outcome, ***$p < 0.0001$, $n = 71$ cells), Type III (**i**, ITI, $p = 0.2222$; Cue, $p = 0.0525$; Outcome, **$p = 0.0037$, $n = 22$ cells), Type IV (**j**, ITI, ***$p < 0.0001$; Cue, ***$p < 0.0001$; Outcome, ***$p < 0.0001$, $n = 76$ cells), and Type V (**k**, ITI, $p = 0.3169$; Cue, $p = 0.0882$; Outcome, ***$p < 0.0001$, $n = 46$ cells). Data are presented as mean ± SEM. In the box plots, the center line denotes the median, the box boundaries mark the interquartile range and the whiskers extend to the 10th to 90th percentiles.

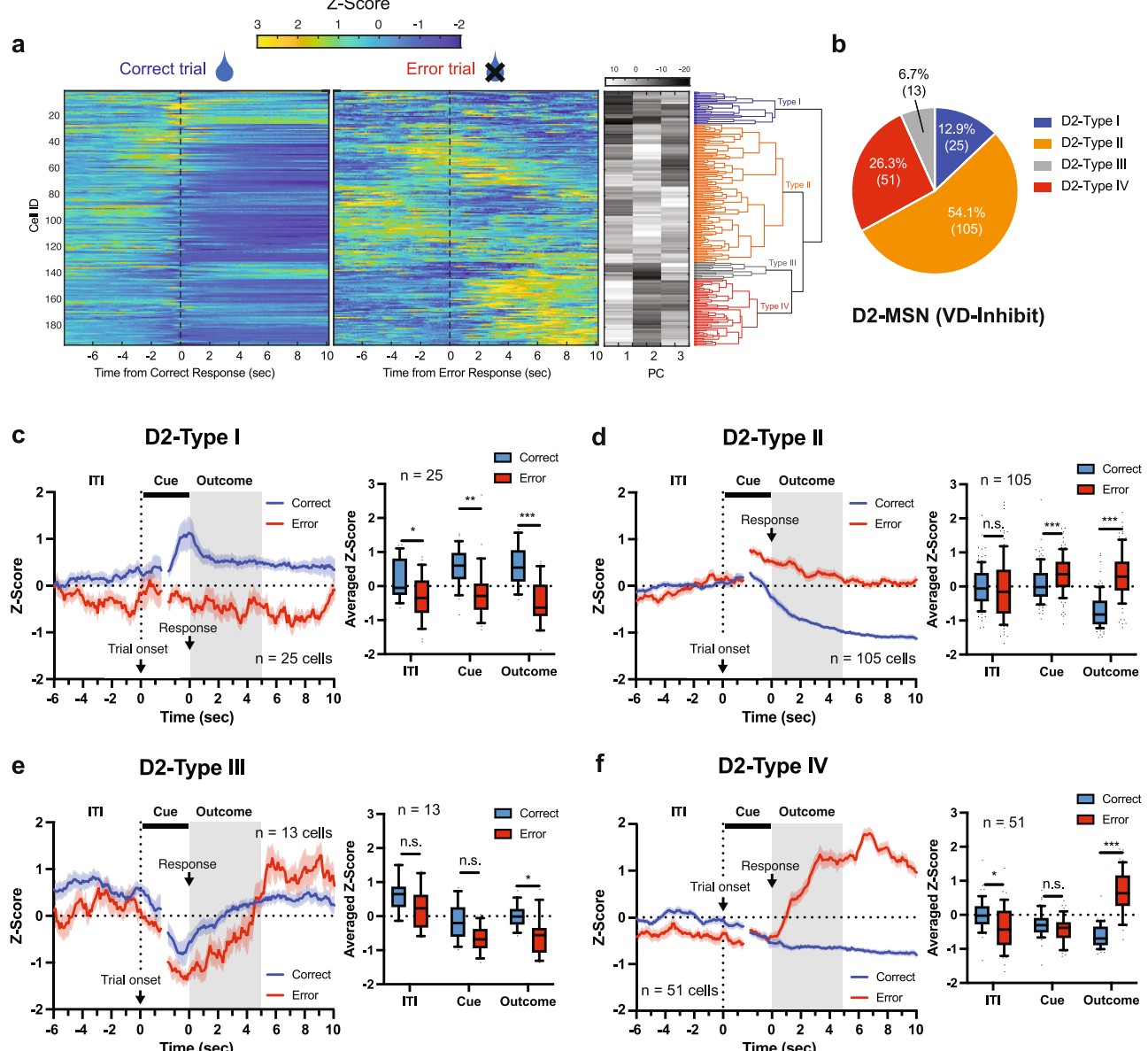

**Fig. 3 | Individual D2-MSNs are predominantly activated by response errors. a** Heatmap of neuronal activity of all D2 neurons recorded in the correct and error trials (Left). Each row represents trial-averaged calcium traces from one neuron. First three principal components (PC) and hierarchical clustering dendrogram showing the relationship of each neuron within the four clusters (Right). **b** Proportion of each type of cell in D2-MSNs. **c**–**f** Population averaged traces of D2-MSNs cell types in correct and error trials (Left). Averaged Z-score during ITI (−5–0 s from trial onset), Cue (0–1.5 s from trial onset and −1.5–0 s from a response), and Outcome (0–5 s from response) period (Right; Two-way

RM ANOVA with Sidak correction) for Type I (**c**, ITI, *$p = 0.0224$; Cue, **$p = 0.0033$; Outcome, ***$p < 0.0001$, $n = 25$ cells), Type II (**d**, ITI, $p = 0.8931$; Cue, ***$p = 0.0002$; Outcome, ***$p < 0.0001$, $n = 105$ cells), Type III (**e**, ITI, $p = 0.2104$; Cue, $p = 0.0593$; Outcome, *$p = 0.0260$, $n = 13$ cells), and Type IV (**f**, ITI, *$p = 0.0121$; Cue, $p = 0.1785$; Outcome, ***$p < 0.0001$, $n = 51$ cells). Data are presented as mean ± SEM. In the box plots, the center line denotes the median, the box boundaries mark the interquartile range and the whiskers extend to the 10th to 90th percentiles.

difference in neural activity between correct and error trials during the Outcome period (Fig. S8c–S8g). Additionally, Type II and Type IV neurons showed a significant difference in activity between correct and error trials during the Cue period (Fig. S8d and S8f).

Four groups of D2-MSNs were identified (Fig. S9b); Type I (33.3%, 39 of 117 cells), Type II (7.7%, 9 of 117 cells), Type III (19.7%, 23 of 117 cells), and Type IV (39.3%, 46 of 114 cells). Here too, large changes in activity were observed within the Outcome period, with two groups of D2-MSNs showing a significant difference in activity between correct and error trials during this time window, Types I (Fig. S9c) and IV (Fig. S9f). With the exception of a significant difference in neural activity between correct and error trials during the ITI period in D2-MSN Type I and Type II neurons (Fig. S9c and S9d), no other significant differences in neural activity between the two trial types were observed in any other task windows (Fig. S9c–S9f).

Taken together with the results of the VD-Inhibit task, our imaging data indicate that NAc D1- and D2-MSNs display a diverse range of activity changes during correct and error trials of the VD-Inhibit and VD-Attend tasks. Most prominently, almost all NAc D1- and D2-MSNs were found to significantly differ in their activity profiles during the Outcome period of correct and error trials, although a smaller amount of neuron types also demonstrated differences in activity between correct and error trials within ITI and Cue periods.

## Outcome-induced signaling in NAc D1-/D2-MSNs was strengthened by learning

Next, to investigate whether differences in the activity patterns NAc D1- and D2-MSNs during the Outcome period are shaped by learning, we analyzed changes in the responses of individual NAc MSNs pre- and post-acquisition of the VD-Inhibit task (Fig. 4a and S3) using a previously-established cell registration method[25]. A total of 239 neurons (D1-Cre, 103 pairs; D2-Cre, 136 pairs) were able to be identified in both the second (Early) and criterion (Late) sessions. The neural activity of cells identified in both Early and Late sessions (registered cells) well matched that of all cells identified in the Late session (originally clustered cells), except for D1-Type III which was removed from subsequent analyses (Fig. S10). We first calculated the degree to which activity in NAc D1- and D2-MSNs during the ITI (−5–0 s from trial initiation), Cue (Cue onset, 0–1.5 s from trial initiation; and Cue offset, −1.5–0 s from a response), and Outcome period (0–5 s from a response) of correct and error trials was altered by learning. In D1-MSNs, cells that will be Type I and Type II in the Late session showed significantly increased and decreased patterns of neural activity during the Outcome period of correct and error trials, respectively (Fig. 4b, c). On the other hand, cells that will be Type IV showed reduced activity during the Cue and Outcome periods and increased activity during the ITI period of correct and error trials, respectively, in late sessions compared with early sessions (Fig. 4d). No statistical difference in neural activity was detected in cells that will be Type V before and after learning (Fig. 4e). The same analysis for D2-MSNs revealed that cells that will be Type I showed increased activity during ITI, Cue, and Outcome periods of correct trials and reduced activity in ITI and Outcome periods of error trials following learning (Fig. 5a); cells that will be Type II became more inhibited during the Outcome period of correct trials and strongly activated during the Outcome period of error trials (Fig. 5b); cells that will be Type IV showed selectively enhanced activation during the Outcome period of error trials (Fig. 5d); and cells that will be Type III demonstrated no significant changes in activity following learning (Fig. 5c).

Next, to analyze how cluster types change before and after learning, hierarchical clustering was performed based on neural activity in the early stages of learning (Fig. S11 and S12). In D1-MSNs, a total of four groups were identified, Type I (17.2%, 17 of 99 cells), Type II (25.2%, 25 of 99 cells), Type III (26.3%, 26 of 99 cells), and Type IV (31.3%, 31 of 99 cells). D1-Type I-Early neurons showed increased

activity during the Outcome period of correct trials compared with error trials (Fig. S11a), while D1-Type III-Early neurons showed the opposite pattern (increased activity during the Outcome period of error trials compared with correct trials) (Fig. S11c). D1-Type II-Early neurons showed increased activity during ITI and Outcome periods of error trials compared with correct trials (Fig. S11b), while D1-Type IV-Early neurons showed increased activity during the ITI period of correct trials compared with error trials (Fig. S11d). Type I-Early and Type II-Early neurons transitioned to Type I, II, IV, and V future clusters at broadly similar rates; whereas, Type III-Early and Type IV-Early transitioned to each of the 5 future cluster types, but with Type IV-Early neurons most likely to transition into future Type IV neurons (those activated during the Outcome period of error trials) (Fig. S11f). In D2-MSNs, a total of four groups were also identified, Type I (13.2%, 18 of 136 cells), Type II (9.6%, 13 of 136 cells), Type III (50.0%, 68 of 136 cells), and Type IV (27.2%, 37 of 136 cells). Type I-Early and Type II-Early neurons showed increased activity during the Outcome period of correct trials compared with error trials (Fig. S12a and S12b). Type III-Early neurons had increased activity during ITI and Outcome periods of error trials than in correct trials (Fig. S12c), and Type IV-Early had higher neural activity in ITI, Cue, and Outcome periods of error trials than in correct trials (Fig. S12d). All D2-MSN-Early neuron types were most likely to transition to Type II future clusters, which are activated during the Outcome periods of error trials (Fig. S12f).

Overall, our in-vivo imaging findings suggest that the activity of D1- and D2-MSNs signaling correct or error responses during the Outcome period appears to be strengthened by repeated training, and may contribute to the concurrent improvement in task performance.

## Optogenetic suppression of NAc D2-MSNs during the outcome period of error trials impairs inhibition-based choice behavior

Our in-vivo calcium imaging experiments demonstrated a diverse range of responses by NAc D1- and D2-MSNs during ITI, Cue, and Outcome periods of correct and error trials in the VD-Inhibit task. However, it was still unclear how this activity functionally contributed to the performance of the task. Therefore, we next sought to optogenetically suppress the activity of NAc D1- and D2-MSNs in a time-locked manner during performance of the VD-Inhibit task. We bilaterally injected a Cre-dependent AAV expressing the light-driven outward proton pump, archaerhodopsin (ArchT)[26], fused to the fluorescent marker tdTomato (AAV5-FLEX-ArchT3.0-tdTomato), or an eYFP marker (AAV5-DIO-eYFP) for control animals, into the NAc of D2-Cre, A2a-Cre, or D1-Cre mice, then implanted optic fibers directly above the dorsolateral NAc (Figs. 6a, b, and S13). This technique has previously been established to suppress the activity of ArchT-expressing NAc D2-MSNs[27]. Once animals had reached criterion levels of responding on the VD-Inhibit task (≥80% correct on two consecutive days or ≥75% correct on three consecutive days), we suppressed the activity of NAc neurons during 3 time periods of the task (ITI period; the last 5 sec of the inter-trial interval (ITI), Cue period; the time from trial initiation to the response, Outcome period; 5 s after the response, Fig. 6c) in separate sessions on consecutive days in a pseudo-randomized (latin-square design) order and observed the effect on task performance (Fig. S14). During each test session, LED stimulation was performed in a pseudo-random (never more than 3 consecutive trials of the same trial time) order in 50% of trials. Additionally, as previous studies have indicated that NAc D2-MSNs in particular play an important role in behavior modification following reward omission, we investigated whether suppression of activity affected performance in trials immediately following correct or error responses[14,28–30]. Optogenetic suppression of NAc D1-MSNs during Outcome periods did not affect performance in the VD-Inhibit task (Fig. 6d). On the other hand, we found that optogenetic suppression of NAc D2-MSNs during the Outcome period impaired performance in trials immediately following a response error in the VD-Inhibit task in

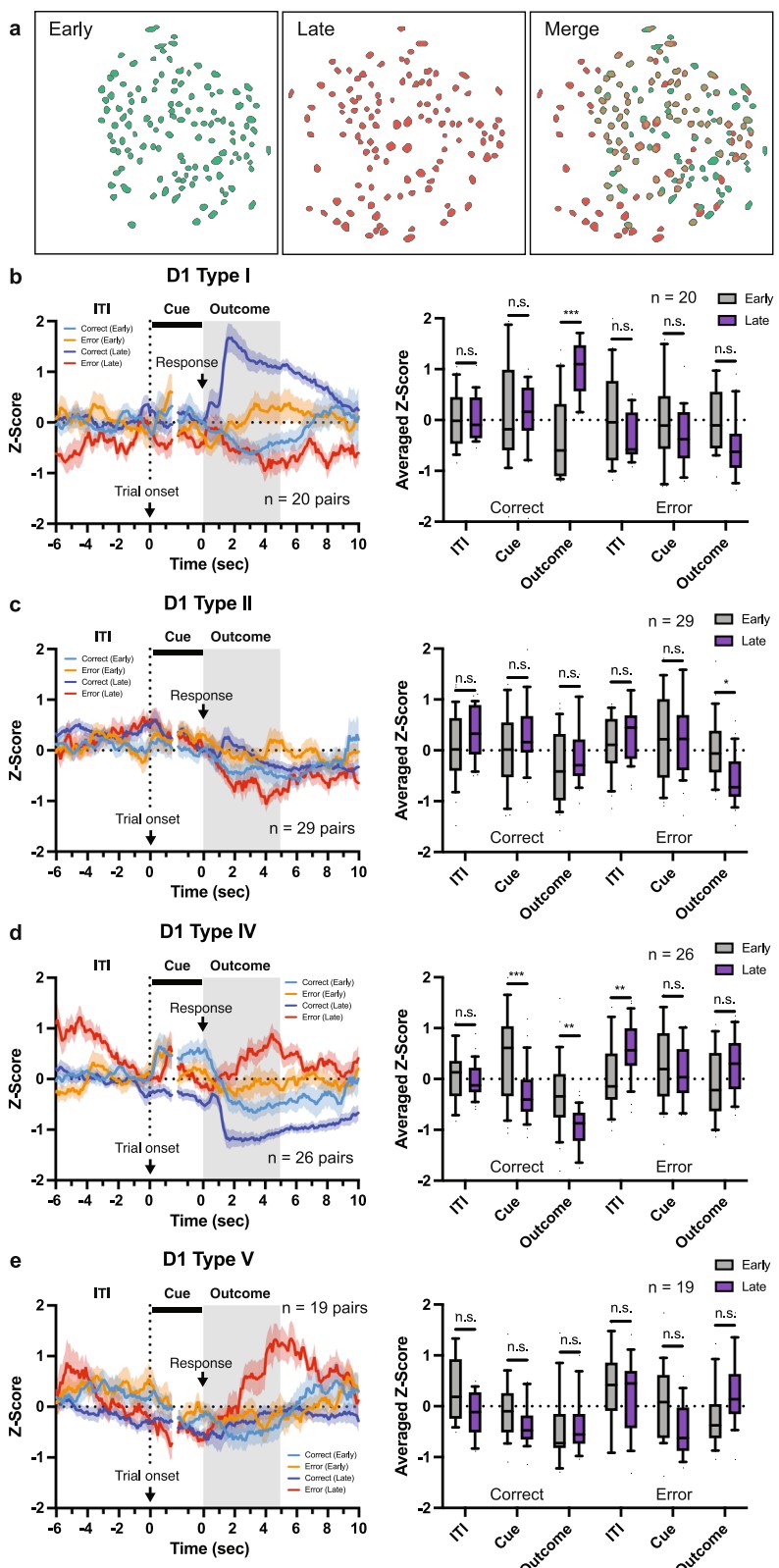

both D2-Cre (Fig. 6e) and A2a-Cre mice (Fig. S15). As expected, no behavioral changes due to LED stimulation were observed in the eYFP-expressing control group (Fig. 6f). Interestingly, while optogenetic suppression of NAc D1-MSNs during the ITI period impaired performance on trials after error responses in the VD-Inhibit task (Fig. S16b), optogenetic suppression of NAc D1-MSNs during the Cue period improved performance on trials after error responses in the VD-Inhibit

task (Fig. S16c). On the other hand, optogenetic suppression of NAc D2-MSNs during the ITI and Cue periods did not affect performance in the VD-Inhibit task (Fig. S16d and S16e). In addition, analysis of performance in the trials following optogenetic suppression revealed that the effect of suppression during ITI and Cue periods on performance was limited to trial in which suppression was performed (Fig. S17). Response latencies and the number of rewards earned, indices of

**Fig. 4 | Activity patterns of D1-MSNs are shaped by learning during different time windows of correct and error trials. a** Representative image of contour map from early (green) and late (red) stages of learning. **b**–**e** Population averaged calcium traces of early (second session) and late (criterion session) sessions in correct and error trials (Left). Averaged Z-score during ITI (−5–0 s from trial onset), Cue (0–1.5 s from trial onset and −1.5–0 s from a response), and Outcome (0–5 s from response) period (Right; Two-way RM ANOVA with Sidak correction) for Type I (**b**, Correct-ITI, $p > 0.9999$; Cue, $p > 0.9999$; Outcome, ***$p < 0.0001$; Error-ITI, $p = 0.3631$; Cue, $p = 0.5017$; Outcome, $p = 0.1323$, $n = 20$ pairs), Type II (**c**, Correct-ITI, $p = 0.5275$; Cue, $p = 0.6013$; Outcome, $p = 0.5903$; Error-ITI, $p = 0.8930$; Cue, $p > 0.9999$; Outcome, *$p = 0.0107$, $n = 29$ pairs), Type IV (**d**, Correct-ITI, $p = 0.9846$; Cue, ***$p = 0.0001$; Outcome, **$p = 0.0011$; Error-ITI, **$p = 0.0073$; Cue, $p = 0.9632$; Outcome, $p = 0.0865$, $n = 26$ pairs), and Type V (**e**, Correct-ITI, $p = 0.1734$; Cue, $p = 0.7418$; Outcome, $p = 0.9972$; Error-ITI, $p = 0.9079$; Cue, $p = 0.0925$; Outcome, $p = 0.1223$, $n = 19$ pairs). Data are presented as mean ± SEM. In the box plots, the center line denotes the median, the box boundaries mark the interquartile range and the whiskers extend to the 10th to 90th percentiles. Clustering types were defined by criterion session data.

motivation, did not differ as a result of LED stimulation at any of the time windows (ITI, Cue, Outcome) (Fig. S18 and S19). Also, the number of trials for each condition (After Correct/ Error and OFF/ON) was comparable between D1- and D2-ArchT groups and the control group (Fig. S20).

### Post-error activation of D2-MSNs is dispensable for attendance-based choice behavior

Finally, we examined whether the post-error outcome-selective contribution of D2-MSNs observed in the VD-Inhibit task (Fig. 6, S15 and S16) is selective to tasks guided by response inhibition, or rather whether it occurs in a task-independent manner, by investigating the effect of the same optogenetic protocol on performance in the VD-Attend task. As in the previous experiment, the tests were performed in animals that had reached the criterion (≥80% correct on two consecutive days or ≥75% correct on three consecutive days). In the VD-Attend task, optogenetic suppression of NAc D1- or D2-MSNs during ITI, Cue, or Outcome period did not affect performance in trials immediately following either correct or error trials (Fig. S21c–S21k). These findings indicate that activation of NAc D2-MSNs following the Outcome period of response errors contributes to the choice behavior in a context-dependent manner, and is not necessary for effective performance of attendance-based choice behavior.

## Discussion

The NAc is a key component of basal ganglia and is thought to contribute to reward evaluation and motivation control by integrating glutamatergic and dopaminergic inputs from the cerebral cortex and ventral tegmental area (VTA), respectively[5,10,31–33]. However, while the importance of the NAc has been discussed in several models of behavioral control[7,8,10,34,35], a precise understanding of the role of the NAc, and its constituent cell-types, in choice behavior has remained elusive. Here we established a visual discrimination task in mice to assess the neural mechanisms underlying cue-guided inhibition-based choice behavior without the influence of reward-associated cues and revealed that this form of choice behavior was controlled specifically by activity in NAc D2-MSNs. A large population of NAc D2-MSNs (D2 Type IV) was found to be activated during the Outcome period of error trials, and signaling in NAc D2-MSNs during the Outcome period of error trials was demonstrated to play an important role in inhibition of inappropriate behavior in the immediate future. In contrast, while populations of NAc D1-MSNs were found to be activated during the Outcome period of correct responses (D1 Type I) and error responses (D1 Type IV and V), optogenetic stimulation of NAc D1-MSNs during this period did not alter animals' ability to perform the task, indicating that this outcome-specific activation does not contribute to inhibition-based choice behavior. Our findings indicate that error-related activity in a subpopulation of NAc D2-MSNs acts to guide future choice behavior by biasing mice away from environmental cues associated with incorrect choice behavior.

The dominance of NAc activity to the Outcome period of the task in our study suggests that D2-MSNs may be important for monitoring and updating of choice behavior, rather than for action selection itself. Interestingly, previous evidence has suggested that optogenetic stimulation of the dorsomedial striatum (DMS) during Cue presentation biases action selection[29], supporting models proposing the DMS to act as an actor and NAc to act as a critic in action selection and action evaluation, respectively[36,37]. The NAc forms part of a limbic processing loop that has been reported to converge with associative/cognitive and motor processing loops, involving the DMS and DLS, respectively, at the level of the SNr[38,39]. This circuitry provides a mechanism through which information about action values, provided by the limbic loop, can be integrated with information about current goals, mediated by the associative/cognitive loop, to dynamically control choice behavior, as has been suggested by recent computational models[10,40].

Interestingly, in the current study, optogenetic suppression of NAc D1-MSN activity during the ITI period decreased the performance in the VD-Inhibit. Given that optogenetic suppression of NAc D2-MSN activity during the ITI period had no effect on performance, it appears that D1- and D2-MSNs of the NAc contribute to performance of the task by activity during different time windows. Moreover, optogenetic suppression of D1- and D2-MSNs during ITI or outcome periods, respectively, decreased performance in trials after errors, suggesting that both D1- and D2-MSNs contribute to updating the action value. In fact, D1-Type IV neurons had higher neural activity in the ITI period following a response error (error trial), indicating that this activity may be important for updating error information. It is also interesting that optogenetic suppression of D1-MSN activity during the Cue period improved performance after error responses. Previous evidence has indicated D1-MSNs to play an important role in associative learning and approach behavior to conditioned stimuli[41], therefore, it is possible that optogenetic suppression of D1-MSNs may have improved performance in trials following response errors by inhibiting approaches to visual cues associated with response errors (in this case, touches to the flag stimulus). Regarding the D1-MSN activity during the Outcome period, our in-vivo imaging analysis indicated that a large population (D1 Type I) of NAc D1-MSNs was activated during the Outcome period of correct trials, while two smaller populations (D1 Type IV and V) were activated during the Outcome period of error trials. It is possible that optogenetic inhibition of NAc D1-MSNs as a whole population resulted in a canceling out of the functional effects of these opposing populations, neutralizing their influence on performance in the task. An alternative hypothesis is that changes in the activity of NAc D1-MSNs during the task, particularly the Outcome period, may reflect the signaling of salient events, rather than signaling outcome value, which when silenced does not adversely affect task performance. Indeed, an important role for the NAc core in salience signaling has previously been reported[42].

The identification of a large population of NAc D1-MSNs (D1 Type II) that were inhibited during the Outcome period of the VD-Inhibit task may indicate that suppression of NAc D1-MSNs also contributes to error signaling. While optogenetic inhibition of NAc D1-MSNs during the Outcome period of error trials did not significantly improve performance in the VD-Inhibit task, it is possible that error signaling had already reached a ceiling beyond which its strengthening has little effect. Optogenetic activation of D1-MSNs during the Outcome period VD-Inhibit task may help to clarify whether this inhibition is necessary for error signaling, although, here too, whole population stimulation of NAc D1-MSNs may lead to diluted effects as optogenetic activation of the NAc has been reported to produce transient reward[43,44]. Future

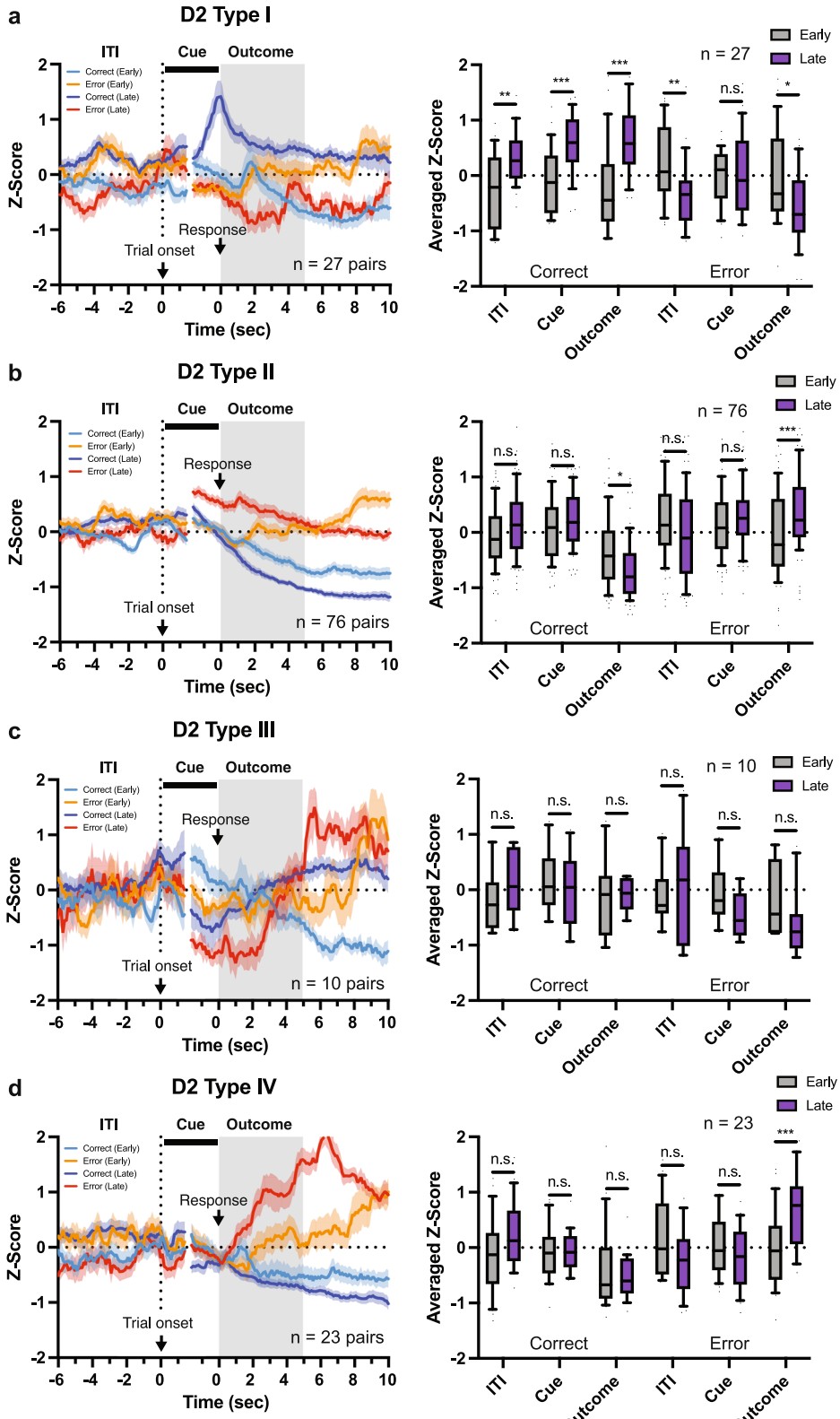

**Fig. 5 | Neural activity patterns of D2-MSNs are strengthened by learning.**
**a–d** Population averaged calcium traces of early (second session) and late (criterion session) sessions in correct and error trials (Left). Averaged Z-score during ITI (−5–0 s from trial onset), Cue (0–1.5 s from trial onset and −1.5–0 s from a response), and Outcome (0–5 s from response) period (Right; Two-way RM ANOVA with Sidak correction) for Type I (**a**, Correct-ITI, **p = 0.0087; Cue, ***p = 0.0004; Outcome, ***p < 0.0001; Error-ITI, **p = 0.0096; Cue, p = 0.9996; Outcome, *p = 0.0128, n = 20 pairs), Type II (**b**, Correct-ITI, p = 0.0653; Cue, p = 0.7686; Outcome, *p = 0.0108; Error-ITI, p = 0.0833; Cue, p = 0.8356; Outcome, ***p = 0.0008,

n = 76 pairs), Type III (**c**, Correct-ITI, p = 0.7481; Cue, p = 0.9871; Outcome, p = 0.9992; Error-ITI, p = 0.9543; Cue, p = 0.4694; Outcome, p = 0.4020, n = 10 pairs), and Type IV (**d**, Correct-ITI, p = 0.1516; Cue, p = 0.9999; Outcome, p = 0.8574; Error-ITI, p = 0.0905; Cue, p = 0.7638; Outcome, ***p = 0.0009, n = 23 pairs). Data are presented as mean ± SEM. In the box plots, the center line denotes the median, the box boundaries mark the interquartile range and the whiskers extend to the 10th to 90th percentiles. Clustering types were defined by criterion session data.

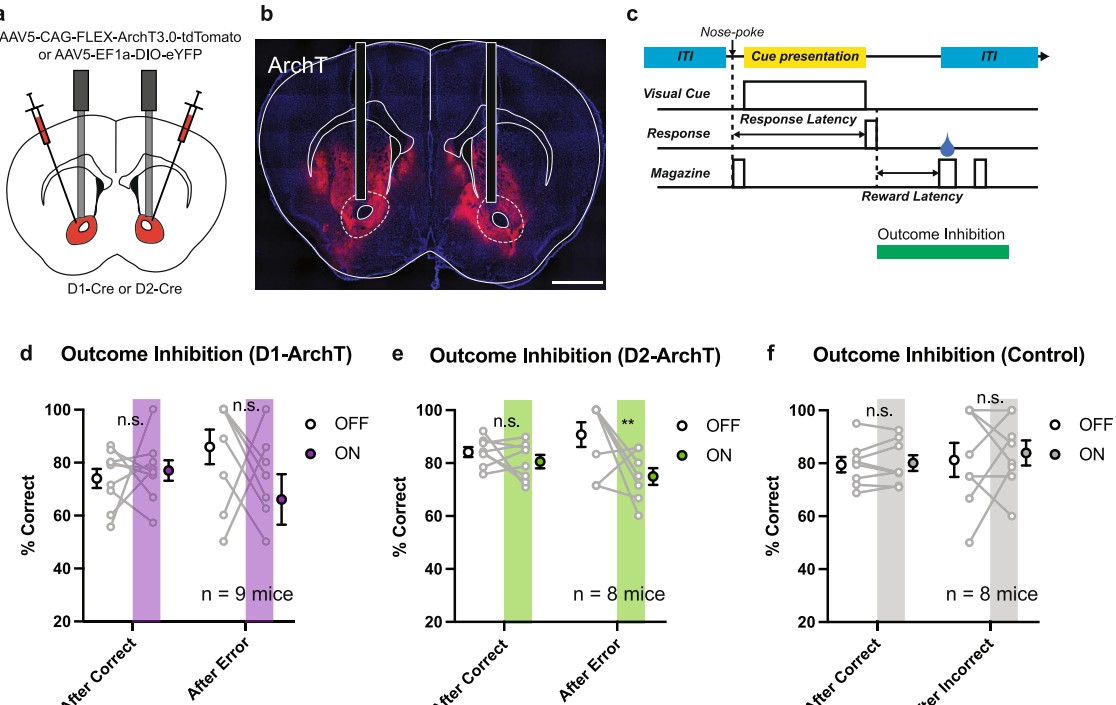

**Fig. 6 | Post-error activation of D2-MSNs is necessary for inhibition-based choice behavior. a** Schematic of viral injection and optic fiber implantation. **b** Representative coronal section with optic fibers. Scale bar, 1 mm. **c** Schematic of optical stimulation protocol. **d** Optogenetic suppression of D1-MSNs in the NAc during the Outcome period did not affect the performance of ArchT mice in the next trial (Two-way RM-ANOVA with Sidak correction; After Correct, $p = 0.9890$; After Error, $p = 0.6913$, $n = 9$ mice). **e** Optogenetic suppression of D2-MSNs in the NAc during the Outcome period following a response error decreased the performance of ArchT mice in the next trial (Two-way RM-ANOVA with Sidak correction; After Correct, $p = 0.5716$; After Error, **$p = 0.0061$, $n = 8$ mice). **f** LED delivery to the NAc during the Outcome period did not affect the performance of control (eYFP) mice in the next trial (two-way RM-ANOVA with Sidak correction; After Correct, $p = 0.8660$; After Error, $p = 0.8660$, $n = 8$ mice). Data are presented as mean ± SEM.

studies using transectional viral expression techniques that allow the tagging of specific cell types activated at precise time points, such as those used in recent studies[45,46], may allow for more precise identification of the functional contributions of subpopulations of NAc D1- and D2-MSNs to error signaling and choice behavior.

An interesting finding of our in-vivo imaging data was that while populations of NAc D2-MSNs activated by the negative outcome (reward omission) of a response error remained largely consistent throughout task learning, populations of NAc D1-MSNs activated by the rewarding outcome (reward delivery) of a correct response changed across task learning. A potential explanation for these activity patterns may be that NAc signaling does not simply encode the value of a specific outcome, but rather encodes more complex information about the value of outcomes associated with specific cues. Thus, it could be speculated that the random nature of the reward-associated cue, but not the non-reward-associated cue, in our inhibition-based visual discrimination (VD-Inhibit) task resulted in the changing patterns of activated NAc D1-MSNs, but not NAc D2-MSNs. Alternatively, it is possible that rewarding outcomes are signaled by more general, summated patterns of activity than negative outcomes. Indeed, recent evidence suggests that NAc D1-MSN activity controls generalized Pavlovian learning of cue-outcome associations, while NAc D2-MSNs contribute to the ability to discriminate between Pavlovian cues[5]. A further surprising finding of our in-vivo imaging study was that, contrary to the canonical role of NAc D1-MSNs in reward signaling[13,43,44,47,48], a small subset of NAc D1-MSNs were found to be activated by the reward omission. These data hint at heterogeneous functionality of NAc D1-MSN subpopulations and suggest the potential existence of NAc D1-MSNs responsive to negative outcomes, as has recently been reported in dorsal striatal D1-MSNs[49].

Previous studies employing electrophysiological recording of the NAc show that the majority of the NAc neurons are activated or inhibited in response to the reward itself[50–52]. Although it is not easy to directly compare these findings with our own due to differences in the task designs used, the pattern of rapid activation in response to rewards in our study (D1-Type I and D2-Type I) is in agreement with the electrophysiological characteristics of NAc neurons previous identified. Conversely, we observed a certain number of cells that are suppressed for a long time after the reward is acquired (D1-Type IV and D2-Type II). This difference might be due to the fact that the calcium sensor GCaMP also captures changes in intracellular signaling[53]. Compared to previous studies, a smaller percentage of cells responded to the onset of cue presentation, but this may be because the nature of the task did not allow for immediate confirmation of cue, and the response to cue may have been mixed with motor preparation activity. Moreover, it is also important to note differences in functionality across different subregions of the NAc. Optogenetic stimulation of D1-MSN in the lateral NAc and dynorphin neurons (most dynorphin expressing neurons coexpress the D1 receptor[54,55]) in the dorsal NAc is rewarding, whereas optogenetic stimulation of D1-MSN in the medial NAc and dynorphin neurons in the ventral NAc causes general behavioral inhibition and aversion, respectively[56,57]. In the present experiment, we targeted the dorsolateral region of the NAc (Fig. S2 and S8), and the fact that many D1-MSNs responded positively to the reward is in agreement with the findings of previous studies. On the other hand, several recent studies have suggested that D1- and D2-MSNs in the NAc core and shell may cooperatively contribute to reward and aversion signaling, motivation, and behavioral inhibition[44,58–60]. Given that the dynamics of DA in NAc core and shell regions in response to electric shocks are opposite[61], one would expect D2-MSNs to function differently in the core and shell region. Future studies investigating the

precise activity patterns of populations of NAc D1- and D2-MSNs in response to a variety of rewarding and negative stimuli will likely help to fully elucidate the roles of these neural populations in value signaling.

The NAc receives dense projections from dopamine (DA) neurons of the VTA, which are known to encode reward prediction error signals[35,62]. Upon encounter of a reward greater than that predicted from previous experience, VTA DA neuron activity, and local DA release in the NAc core, are reported to increase, while the opposite pattern is observed when a reward smaller that predicted is encountered[5,35,61]. Within the NAc, the excitability of D1- and D2-MSNs are likely to be bidirectly modulated by local DA release[63,64]. Indeed, while DA binding at Gs-protein coupled D1 receptors stimulates cAMP signaling, increasing cellular excitability, binding at Gi-protein coupled D2 receptors inhibits cAMP signaling, reducing the cell's excitability[65]. Thus, the activity of NAc D1- and D2-MSNs observed in the present study largely corresponds with that expected according to the reward prediction error theory and the molecular properties of DA receptors; with a rewarding outcome predominantly activating NAc D1-MSNs, likely as a result of augmented local DA release, and a negative outcome predominantly activating NAc D2-MSNs, likely as the disin-hibitory effect of a reduction in local DA release. These findings also support previous evidence indicating that NAc D1- and D2-MSNs play important roles in signaling reward and aversion, respectively[13,43,48,66,67]. Interestingly, studies in humans have shown that increased DA concentration in the brain following L-dopa treatment, reduces the ability of participants to inhibit choices that lead to negative outcomes, but does not alter the ability to learn from positive outcomes in goal-directed learning tasks[68,69]. Given that optogenetic suppression of NAc D2-MSNs in our study similarly disrupted the ability of mice to inhibit a behavioral response leading to a negative outcome when error-signals were blocked, but not when reward signals were blocked, it could be speculated that the ability of L-dopa treatment to impair inhibition-based choice behavior in humans may have been the result of DA-induced hypoactivity of NAc D2-MSNs.

Finally, impairments in the ability for response control are associated with risk-taking behaviors in drug addiction and attention-deficit/ hyperactivity disorder[69–72]. A previous study showed D2-MSN activation in the NAc decreased risky choices in the risk-seeking rats, suggesting that D2-MSN activity in the NAc is important for inhibiting risk-taking behavior[73]. Our results largely support their data and extend them to show that NAc D2-MSN activity is involved in inhibiting the action associated with the negative outcome as well as a risky choice. Additionally, their data showing that the probability of making a risky choice decreases on the next trial after failing to obtain a reward and that optogenetic activation of NAc D2-MSNs reduces risky behavior are in agreement with our findings. Repeated cocaine treatment has also been shown to reduce the frequency of miniature excitatory postsynaptic currents in D2-MSNs[74]. This study fits with a model in which increased excitability of D2-MSNs leads to the strategy to inhibit a bad option, while decrease in excitability of D2-MSN causes to disability to inhibit a bad option[47,69,75]. These bidirectional effects on the strategy support our hypothesis that activation of D2-MSN plays an important role in inhibiting a bad option.

In conclusion, we provide evidence that activation of D2-MSNs in the dorsolateral NAc by response errors plays an important role in the ability to use environmental cues to guide inhibition of undesirable responses. These findings indicate that modulating the neural activity of D2-MSN in the NAc by D2 receptor-selective drugs may be beneficial for the treatment of disorders associated with impaired ability for inhibitory control, such as drug addiction and attention-deficit/ hyperactivity disorder[69–72]. In addition, our findings suggest that the VD-Inhibit task is a useful paradigm for investigating the neural mechanisms that underlie inhibition-based choice behavior.

## Methods

### Animals

Wild-type C57BL/6J mice (male, 8–10 weeks old) were used for validation of behavioral experiments. For optogenetic, and calcium imaging experiments, male heterozygous D1-Cre (FK150Gsat), D2-Cre (ER44Gsat), and A2a-Cre 2 M strain mice were used (optogenetics, D1-Cre, $n = 9$; D2-Cre, $n = 19$, 2 mice were excluded due to insufficient conditioning; A2a-Cre, $n = 19$; calcium imaging, D1-Cre, $n = 3$; D2-Cre, $n = 4$, one mouse was excluded because of incorrect GRIN lens placement). D1-Cre, D2-Cre, A2a-Cre were maintained in a C57BL/6J background. Animals were kept at 23 °C and 55% humidity on a 12-h light/ dark cycle. Behavioral studies were conducted during the light cycle. Mice were kept on water restriction during behavioral testing. For all behavioral experiments except for calcium imaging experiment, mice were grouped housed throughout the experiments. For calcium imaging experiments, mice were singly housed after GRIN lens implantation. All experiments conformed to the guidelines of the National Institutes of Health experimental procedures, and were approved by the Animal Experimental Committee of Institute for Protein Research at Osaka University (approval ID 29-02-1 and R04-01-0).

### Stereotaxic surgery

All mice used in this study were anesthetized with ketamine (100 mg/ kg) and xylazine (20 mg/kg) for surgical procedures and placed in a stereotaxic frame (Kopf Instruments, CA, USA).

For optogenetics experiments, heterozygous D2-Cre mice were bilaterally injected with 400 nl of AAV5-CAG-FLEX-ArchT3.0-tdTomato ($1.3 \times 10^{13}$ GC/ml, Addgene, MA, USA) or AAV5-EF1a-DIO-eYFP ($1.3 \times 10^{13}$ GC/ml, Addgene, MA, USA) using a Nanoject III instrument (Drummond, PA, USA) at a rate of 100 nl/min (coordinates in mm: AP + 1.20, ML ± 1.25 from bregma, and DV − 3.50 from brain surface. The injection pipette remained in place for 5–10 min to reduce backflow. After retraction, 200 μm diameter (NA 0.37) optic fibers (Thorlabs, NJ, USA) were bilaterally implanted and fixed in place with the dental cement (Superbond; Sunmedical, Shiga, Japan) at AP + 1.20, ML ± 1.30 from bregma, and DV − 3.20 from brain surface.

For calcium imaging experiments, AAV9-FLEX-jGCaMP7f ($3.8 \times 10^{13}$ GC/ml, Addgene) was diluted 4-fold in saline and stereotaxically injected into heterozygous D1-Cre or D2-Cre mice using a Nanoject III instrument (Drummond, PA, USA) at a rate of 100 nl/min (coordinates in mm: AP + 1.20, ML 1.25 from bregma, and DV − 3.60 and −3.10 from brain surface (600 nl each depth). The injection pipette remained in place for 5–10 min to reduce backflow. After virus injection, a sterile 21-gauge needle was slowly lowered into the brain to a depth of −2.0 mm from the brain surface to aspirate brain tissue above the NAc. A GRIN lens (600 μm diameter, Inscopix, CS, USA) was slowly lowered into the brain to a depth of −3.20 mm from the brain surface by using a GRIN lens holder (Inscopix, CA, USA). We secured the GRIN lens to the skull with dental cement (Superbond; Sunmedical, Shiga, Japan). A silicone elastomer (Kwik-Cast; World Precision Instruments, FL, USA) was applied to the top of the lens to prevent external damage. Four-to-six weeks after lens implantation, a baseplate (Inscopix, CA, USA) attached to the miniature microscope (nVista; Inscopix, CA, USA) was positioned above the GRIN lens. The focal plane was adjusted until blood vessels could be clearly observed. After adjustment, the baseplate was secured in place with the dental cement.

### Behavioral experiments

**Apparatus.** Training and testing were conducted in a Bussey-Saksida touchscreen chamber (Lafayette Instruments, IN, USA). A black plastic mask with 2 windows ($70 \times 75$ mm$^2$ spaced, 5 mm apart, 16 mm above the floor) was placed in front of the touchscreen. ABET II and Whis-kerServer software (Lafayette Instruments, IN, USA) were used to control operant system and data collection.

**Pretraining.** Pretraining was conducted according to a previously described method with minor modifications[4]. Briefly, during the first phase (Habituation sessions: 3 days), mice were habituated to the chamber in three consecutive daily 40-min sessions. Diluted condensed milk (7 μl, Morinaga Milk, Tokyo, Japan) was dispensed in the reward magazine every 10 sec. In the following phase (Must initiate session: criterion = 60 trials completed in 60 min), 1 of 51 random visual stimuli was displayed in 1 of the 2 windows. After a 30-sec stimulus presentation, the milk reward (20 μl) was delivered with a tone (3 kHz) and the inside of the magazine was illuminated. When mice collected the reward, the magazine light was extinguished and the next trial commenced with a new stimulus following a 20-s intertrial interval (ITI). In the next phase (Must touch session: 60 trials completed in 60 min), stimuli were randomly displayed in 1 of the 2 windows, and mice were obligated to touch the stimulus to receive a reward. In the final phase of the pretraining (Punish incorrect sessions: criterion = (>75% trials correct within 35 min for 2 consecutive days), when a blank window was touched, mice were punished with a 5-s time-out in which the house light was illuminated. After reaching criterion, mice moved on to basic training. Mice typically took 1 day to complete the "Must initiate" session, 1 day to complete the "Must touch session", and 2–4 days to complete the "Punish incorrect" session, in line with completion times previously described using a similar pretraining protocol[4].

**Basic training.** Mice were tested 5–6 days per week (60 trials per day, or up to 60 min). Each trial was initiated after mice nose-poked in the reward magazine. Visual cues were presented until mice responded at either window.

For the VD-Attend task, two visual cues (marble and a random image) were presented in the touchscreen. The random image was pseudorandomly chosen from 51 images. If the mouse responded to the correct (marble) visual cue, a milk reward (7 μl) was delivered with a tone (3 kHz) and the magazine was illuminated. When mice collected the reward, the magazine light went out, and the next trial commenced (60 trials, or up to 60 min) with a new stimulus after a 20-s intertrial interval (ITI). If the mouse responded to the incorrect (random) visual cue, the mouse was punished with a 5-s time-out (house light on).

For the VD-Inhibit task, two visual cues (flag and a random image) were presented in the touchscreen. If the mouse responded to the correct (random) visual cue, a milk reward (7 μl) was delivered with a tone (3 kHz) and the magazine was illuminated. If the mouse responded to the incorrect (flag) visual cue, the mouse was punished with a 5-s time-out (house light on).

A response at a random image was recorded as a correct response, while a response to visual cue "Flag" was recorded as an incorrect response.

After reaching criterion (>80% correct for 2 consecutive days), mice moved on to cable habituation for optogenetic or miniature microscope imaging experiments.

For optogenetic suppression experiments, once performance had stabilized (>80% correct for 2 consecutive days) with the fiber optic cables attached, optogenetic stimulation test sessions were performed. Continuous LED stimulation was performed at 1–3 mW using a 550 nm LED attached to a rotary joint (Plexon, TX, USA). LED stimulation was performed during the ITI (−5–0 s from trial onset), Cue (the time from trial initiation to the response), or Outcome (0–5 s from a response) period of the test session, with all mice receiving each stimulation type (ITI, Cue, or Outcome) in separate sessions on consecutive days in a pseudo-randomized order (latin-square design). Additionally, LED stimulation was only performed in 50% of all trials in a pseudo-randomized manner (never more than three consecutive trials of the same type (LED on/off)). VD-Inhibit and VD-Attend tasks were performed in the same individuals, with VD-Inhibit always performed first.

For calcium imaging experiments, data was acquired at 20 Hz with 0.6 mW LED at the second session of the training (Early) and the session after reaching criterion (the criterion session; Late). After acquisition, calcium recording files were temporally (factor of 2) and spatially (factor of 4) downsampled and motion-corrected using Inscopix Data Processing software ver 1.3.0 (Inscopix, CA, USA). The fluorescent traces of individual neurons were extracted from these images by a constrained nonnegative matrix factorization (CNMFe) method[22]. Z-scores were normalized using values from −10 to +10 s from the start of the trial. Neuronal activity patterns were classified based on their activity patterns through a previously described unsupervised clustering approach[62]. Briefly, the first three principal components of the Z-scored neuronal activities of all neurons averaged across all correct and error trials were calculated using principal component analysis (PCA), with the singular value decomposition algorithm. Hierarchical clustering of the first three principal components was then performed using a Euclidean distance metric and a complete agglomeration method. MATLAB's linkage function was used to perform the hierarchical clustering. To track the same cell in the early and late stages of learning, we compared the maximum intensity projection map of Session 2 (early) with that of the criterion session (late) within the same animals[76,77] and registered identical cells using the plug-in function of Inscopix Data Processing software (NCC Score greater than 0.5). VD-Inhibit and VD-Attend were recorded from the same individual, with VD-Inhibit always performed first.

Percentage correct (correct trials divided by correct plus incorrect trials, recorded as percent), and latencies to correct response, incorrect response, and reward collection were monitored in all behavioral experiments.

## Histology
Animals were deeply anesthetized and transcardially perfused with 0.01 M PBS followed by 4% paraformaldehyde (PFA) in 0.1 M PB (pH 7.4). Brains were removed and post-fixed with 4% PFA at 4 °C for 2 days. After cryoprotection, brains were embedded in OCT compound and cryosectioned (40 μm). Sections were mounted with antifade mouting medium with DAPI (Vectashield). Stitched images were acquired using a Keyence BZ-X800 microscope (Keyence, Osaka, Japan).

## Statistics and reproducibility
Prism (Graphpad, CA, USA) software was used for statistical analyses. The Anderson-Darling test was used to confirm the normality of the data. The behavioral performances in wild-type were analyzed using unpaired *t*-test, paired *t*-test, Mann Whitney test, or Wilcoxon signed rank test. Optogenetic data were analyzed using two-way RM ANOVA with Group (ArchT, eYFP) and Light stimulation (OFF, ON) or History (After Correct, After Error) and Light stimulation (OFF, ON). Post hoc Sidak's multiple comparisons test was performed when F-ratios were significant ($p < 0.05$). The Geisser-Greenhouse correction was applied when non-normal distribution was observed. All data are expressed as means ± SEM. All behavioral and imaging data were acquired from a minimum of 2–5 independently performed experimental series. All attempts at replication were successful.

## Reporting summary
Further information on research design is available in the Nature Portfolio Reporting Summary linked to this article.

# Data availability
Source data are provided with this paper.

# Code availability
Codes used to analyze or display the data are available upon request.

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

## Acknowledgements

This work was supported by MEXT/JSPS KAKENHI grants numbers, JP21K15209 (to T.N.), JP21K15210 (to T.M.), JP19H04983 and JP21H02804 (to K.H.), JP16H06568, JP18H02542, JP21H05694, JP22H02944, and JP22H00494 (to T.H.), by JSPS, Overseas Research Fellowships (to T.N.), by AMED under grant numbers JP21wm0425010 and JP21gm1510006 (to T.H.), by Takeda Life Science Research Foundation (to T.H.), by the SENSHIN Medical Foundation (to T.H.), and by the International Collaborative Research Program of Institute for Protein Research, Osaka University, ICRa-22-03 (to T.H.). A.K.E is a research director of the FRS-FNRS and a WELBIO investigator. We thank members of Hikida laboratory for maintenance of animals and helpful discussion.

## Author contributions

T.N. conceived the project, conducted calcium imaging experiments, performed the analysis, and wrote the manuscript. T.N. and S.A. conducted the optogenetic experiments. T.M. supported calcium imaging experiments. M.L. and A.K.E. contributed materials. K.H. contributed to the analysis. T.M. and T.H. supervised the project. T.M., K.H., and T.H. reviewed and edited the manuscript.

## Competing interests

The authors declare no competing interests.
