## [Peer Review File · Nature Communications]

Error-related Signaling in Nucleus Accumbens D2
Receptor-expressing Neurons Guides Avoidance-based Choice
BehaviorREVIEWER COMMENTS

Reviewer #1 (Remarks to the Author):

In this manuscript the authors design an appetitive conditioning task in mice using touchscreen technology in which the animals are trained to diverse associations in order to obtain liquid food rewards. They then combine a series of high-throughput systems neuroscience techniques to study the behaviour of the two main neuronal populations in the nucleus accumbens during this task.

The first and most important problem is that the behavioural task is mischaracterised, based on well-established associative learning phenomena. There are two major issues with it:

(a) This is not an instrumental task. The parameters measured here involve approaching and “touching” a conditioned stimulus (CS), which is a typical Pavlovian conditioned response (CR)—no different from the “pecking” of pigeons on the CS in classic Pavlovian conditioning experiments. While the animals are required to touch the screen in order for the reward to be delivered at some point in training, this still does not define this response as instrumental (e.g. similar strategies driving CRs are typically used to provoke future instrumental action, such as the shaping phenomenon). As is, and without the existence of manipulanda that define instrumental action, this cannot be considered a goal-directed instrumental task.

(b) This is not an avoidance task. This paradigm relies on never pairing the flag with reward. The overall design used is best characterised as A+ X-; B- X+ where A is the marble and B is the Flag. X is a CS on its own which relies on randomly presented stimuli. It is hard to tell what specific associative process drives the responses because of inaccuracies of the design. Seems like the effect observed could rely on conditioned inhibition to the flag since it is not paired with reward, but in any case inhibitors cause withdrawal, not avoidance. Excitation, however, is clearly driven by A+ vs. AX- X- in the marble condition and by B- BX- X+ in the flag case; where, in the former, A (the marble) is a Pavlovian executor and, in the latter, B (the flag) is a Pavlovian inhibitor. Thus, the responses measured in this paradigm entirely depend on Pavlovian discrimination, and have nothing to do with avoidance behaviour. Avoidance is tactically where the response itself is what inhibits an aversive state (fear/frustrative non-reward). Here, what reduces the aversive state signalled by the flag is not well defined and remains ambiguous. Most likely, the force driving the response is the alternative stimulus, which is, of course, associated with reward, and not the response itself. On a related note, the flag and marble stimuli are not counterbalanced across tasks. Why not? Perhaps the flag is not very salient and so is a poor stimulus. If so, that alone would explain the difference in performance observed.

A second potential major issue in this paper is the fact that they use D2-Cre mice. The field has long moved away from using D2R to drive the targeting to D2-expressing MSNs due to the fact that a shorter isoform of the receptor (D2s as opposed to D2l) is also encoded by the same promoter but is strongly expressed in pre-synaptic dopamine terminals. This means, in this case, that Cre is present in dopamine

terminals in the striatum, which can be very problematic, especially considering the rather promiscuous tropism that AAVs have when infecting membranes in the neuropile (where even retrograde transport is not uncommon, e.g. Aschauer et al 2013). Indeed, most data in Figure 2 (which are not immediately consistent with the predictions from the literature that D1- and D2-MSNs oppose each other) may very well be explained by this particular artefact. For example, it is not immediately logical that overall % of correct trials drops in both cases (Figure 2D). The way I'd interpret this, D1-iDREADDs mice drop performance because D1 neurons are inhibited (as intended; makes sense as these neurons go straight to SNr and are considered the "effectors" of the striatum), whereas D2-iDREADDs drop performance because dopamine drops (through DREADDs-mediated terminal inhibition), which consequently under-stimulates D1-MSNs—ultimately resulting in the same effect. Based on this, one could even provide a good alternative explanation for the differences seen between genotypes in performance after correct or after error trials (Fig. 2E and F). There, both D1-iDREADDs and D2-iDREADDs show the same performance drop after correct trials (this could be attributed to D1-MSN inhibition in the former and dopamine dropping in the latter, as above). On the other hand, both manipulations differ in "After Error" performance: D1-iDREADDs remains invariable while D2-iDREADDs drops again. This latter effect could indeed be due to the postsynaptic D2-MSN inhibition intended in the experiment, but it is overshadowed by the fact that "After correct" performance also goes down. To be fair with the conclusions provided here on error-based learning, the D2-iDREADDs group should show unaltered "After Correct" performance and blunted "After Error" performance. An experiment using A2a-Cre mice instead could reconcile these results and support the claim and main conclusion of the paper.

Other Comments

1 – I do not quite see how the post-error slowing effect mentioned in lines 79-84 (Figure S1) is relevant here, especially since the effect is not even significant. I may help removing this part altogether.

2 – It should be acknowledged that both core and shell of the NAc received DREADDs as per Figure S2. The way information in lines 96-99 reads, one understands that only the Core was manipulated.

3 – The rationale for comparing "After correct" and "after error" responses could be better articulated. Perhaps a bit more literature on error-based correction in the literature would have helped highlighting the importance of these analyses.

4 – Related to my main comment above, I don't think that the data presented in Figure S5 supports the claim that "D2-MSNs are implicated in avoidance-based goal-directed behaviour following response errors". Here performance does not even change in "After Error" trials (Fig S5E).

5 – In Figure 3, why did the authors run the clustering analysis in the entire dataset only? It is indeed interesting to see that a general analysis pulled a large percentage of D2 cells as Type II, but I think this is only the beginning. This analysis should perhaps be followed by a second clustering study where D1 and

D2 neurons are analysed separately. It would be very helpful to see the behaviour of all D1 neurons against the behaviour of all D2 neurons side by side in correct and error conditions - the statistical comparison between these datasets would be valid and very informative, particularly for appreciating the heterogeneity of functions that supports one system and the other in correct vs error trials. Furthermore, I think that such separate clustering arrangements would support the analysis conducted in Figure 4 more directly, where the aim is to identify the evolution of activity behaviour in identified neurons of each system throughout learning. As is, all analyses in Figures 3 and 4 are based on a general clustering analysis run on both populations lumped together, which is not immediately logical based on the core hypothesis of this study. To start with, D1-MSN and D2-MSN responses are not even collected from the same transgenic mouse strain: combining these datasets as one could have statistical concerns.

6 – In figure 5G, a significant trial type x light interaction ensuring that the drop of correct responses seen upon inhibition are indeed due to the trial type considered would be necessary (only the simple effect is reported in the figure legend).

Minor comments

1 – Start of line 30: outcome instead of outcomes

2 – What is the "shaping" phase indicated in Fig 2C? I can't find this in the methods.

3 - In Figure 3, I believe the normalised Z-score colorimetric scale under panel F belongs to panel E?

4 – Line 201: “Nevertheless, we detected a significant number of cells that acquired discriminability between correct and error outcomes (Fig. S10) in both cell types.” - I don't seem to be able to find these data in Figure S10...?

5 – Line 245: models instead of model

Reviewer #2 (Remarks to the Author):

In this interesting work, authors used a new visual discrimination task in mice, and assessed the role of NAc D1-/D2-MSNs in cue-guided goal-directed “avoidance”. By performing cell-type specific neuronal silencing and in-vivo imaging, authors propose that NAc D2-MSNs to selectively contribute to cue-guided avoidance behavior, with activation of NAc D2-MSNs following response error playing an important role in optimizing future goal-directed behavior. They further suggest that error-signaling by NAc D2-MSNs underlies the ability to use environmental cues to avoid inappropriate behavior.

The methodology is adequate, the paper is well written and experimental controls are provided. The manuscript is interesting but has some methodological and conceptual issues that need to be solved/clarified and some experiments performed in order to support the claims. Please find below my main concerns regarding the current version of the manuscript. I truly hope that these comments and

suggestions will be useful for the authors and improve the status of the manuscript.

Major comments

1. Is avoidance a correct term? The term “avoid” is usually used to define an adaptive behavior in response to an aversive situation, including active and passive avoidance. In this task, the animal has to inhibit a particular behavior in order to get a reward, not to prevent something aversive, so I think it is not an adequate term to describe the task.
2. Was imaging also performed in the attend task? This would be an interesting comparison to perform and would strengthen authors message.
3. I have several concerns about the neuronal clustering methodology, but also the interpretation of the calcium imaging data (see below)
4. If D1 and D2 neurons have different relevance for correct vs error trial codification and to cue-guided “avoidance” behavior, then their activity should predict the outcome of a particular trial. authors assume this is the case because the majority of type II neurons are D2, but this is not proven directly with the analysis that is provided.
5. Authors should also provide imaging data for D1- and D2- neurons alone (without merging the two populations), and perform clustering if the sample size is sufficient. This would allow to visualize D1 and D2 activity during task performance and to understand how many subpopulations exist of each neuronal subtype.
6. Viral expression in Fig. 5a shows significant spreading to the dorsal striatum, which can limit some of the interpretations of the paper. In addition, for DREADD experiments, a similar amount of virus was also used, and according to sup fig. 2, the transfection did not occur only in NAc core as authors report, but also shell and dorsal st. Since CNO was injected i.p., the behavioral results may derive from inhibition of other brain regions as well.
7. Regarding optogenetic experiments, a key experiment that is missing is the manipulation of D1-MSNs. In addition, activation of D2 (and D1) neurons during the same periods could provide additional information about the role of each population in behavior. And what is the impact of manipulating these populations in different stages of the vd-attend task? I strongly believe that these experiments would strengthen the message that the authors want to convey with this manuscript.
8. How are these results reminiscent of RPE-like?

Methodology

9. Task design – since this manuscript is the first time that the task is performed, authors should provide detailed information:
 - pretraining phase: a stimulus was randomly displayed – what type of stimulus? One of the 52 images described after? what is time out – light on like in the basic training session?; 77% of correct trials is based on what rationale?;
 - How many days comprised each phase – provide distribution of performance so that new users now what to expect. (this is given in 1e but not for VD-Attend for example)
 - Why did you decided to have the lights on during the time out period? Since the light is turned on together with the tone, it will also be a CS.

10. Z-score calculation: standard z-score, i.e., normalized to the signal of whole session or normalized to baseline of each trial? or other period?
11. 1.2ul of viral delivery for GCAMP experiments is a very high amount. Why did you use this quantity?
12. Cell registration methodology should be detailed.
13. Fig S10: auROC methodology should be detailed.
14. Statistics: did authors check for outliers?

Results

15. Results of attend task should also be provided.
16. Lines 119-121: to say that manipulation of MSNs has an effect in motivation, authors would have to perform other type of experiments.
17. Regarding S4, there is increased latency to reward due to D1-MSN inhibition. Did authors controlled for locomotion due to this manipulation?
18. S4: how can you have huge differences in earned rewards in D1 but not D2 animals since both present reduced % of correct due to the manipulation? This is because animals do not go and get the reward? It was not clear to me. If this is the case, then that reward becomes available for the next trial in the lick? To what session is S4 referring to?
19. Are the animals of the attend session others or the same as the avoid task? This is not explicit.
20. S5 e: I do understand that due to variability there are no differences in after error, but the variability is quite remarkable in comparison to previous graphs, showing that likely the manipulation has some effect as well.
21. Not clear how the clustering was performed in fig 3. The description in methods is not clear. Please revise. 3 PCs refer to what?
22. If you perform the hierarchical clustering based on D1 neurons separately from D2 neurons, do you still observe the same clustering pattern?
23. Fig4D: which neuronal traces of this figure are representative of the posterior clusters?
24. Fig4e: you recorded 3 D1- and 3 D2- animals, how many cells per each animal?
25. Fig4F: absolute numbers should also be presented and not only %
26. Fig4G-J it is not easy to identify ITI, cue periods – time stamps in x axis does not match what is said in the legend
27. Fig4G-J – can you also provide the same type of graphs but separating D1- and D2-neurons?
28. How did authors consider the time out periods in error trials in the imaging analysis?
29. Fig4G-J – Authors should also report in the results section (not only in figure legends) that types III and IV also present changes in the ITI period. These findings mean that even in non-relevant periods of the task, the activity of these neurons is already different between correct vs incorrect trials. How to explain this? Can this different activity predict subsequent trial outcome, i.e, correct vs incorrect?
30. Lines 168-170 – inhibition of neurons during a period does not necessarily mean these neurons are not required for the observed outcome, and this is particularly evident regarding GABAergic neurons as nucleus accumbens MSNs. Inhibition of MSNs can in turn disinhibit downstream target regions and promote a specific response. Thus, the assumption that D1 inhibition during error and activation during

correct trials is mostly related to “correct-related signaling” is a biased interpretation of the findings.

31. Fig 5B-E- signal iti, cue, outcome labels in all graphs

32. S10: the auROC was calculated for the whole trial? Not clear

33. Line 201-202 not sure what how you can infer from the data presented in S10 that cells acquire discriminability?

Introduction/Discussion

34. Other relevant studies provided evidence about the role of D1- and D2-MSNs in reward and aversion-related behaviors and should be properly referenced in the adequate sentences (lines 44-47; lines 267 etc): doi: 10.1016/j.neuron.2015.08.019, doi: 10.1038/s41380-019-0484-3, doi:10.1016/j.celrep.2020.02.095, doi.org/10.1371/journal.pone.0207694, DOI: 10.1038/ncomms11829, amongst others.

35. Earlier ephys studies regarding the role of NAc neurons (lines 296-294) should be mentioned such as those from Carelli’s lab. Though they do not segregate between D1 and D2 neurons, these earlier studies show very robustly the role of NAc neurons in this type of processes

Minor comments

36. line 410 typo

37. figures could be improved. For example fig 2 CNO individual dots should be colored according to the condition – grey red and not in the same color as it is now – other option is to include Vehicle CNO in the x axis.

38. Scales of graphs should be consistent for the same type of result

39. Fig 2d bottom refers to which session?

40. Number of animals in sup fig 4?

41. lines 187-194: authors could improve the description of these results, I suggest to be more systematic in the way they present information.

42. Videos – which animals? D1/D2?

43. Mentioning expert mice and novice mice gives the impression that the data are not actually from the same animal across learning phases. Authors could use early vs late learning?

Ana João Rodrigues

REVIEWER COMMENTS

We would like to thank the reviewers for their thoughtful assessments of the manuscript. We have responded to major and minor comments, with our responses in blue. With the help of these comments, we have also made significant improvements to the manuscript, with changes also represented in blue and line numbers given in this document.

Reviewer #1 (Remarks to the Author):

In this manuscript the authors design an appetitive conditioning task in mice using touchscreen technology in which the animals are trained to diverse associations in order to obtain liquid food rewards. They then combine a series of high-throughput systems neuroscience techniques to study the behaviour of the two main neuronal populations in the nucleus accumbens during this task.

The first and most important problem is that the behavioural task is mischaracterised, based on well-established associative learning phenomena. There are two major issues with it:

(a) This is not an instrumental task. The parameters measured here involve approaching and “touching” a conditioned stimulus (CS), which is a typical Pavlovian conditioned response (CR)—no different from the “pecking” of pigeons on the CS in classic Pavlovian conditioning experiments. While the animals are required to touch the screen in order for the reward to be delivered at some point in training, this still does not define this response as instrumental (e.g. similar strategies driving CRs are typically used to provoke future instrumental action, such as the shaping phenomenon). As is, and without the existence of manipulanda that define instrumental action, this cannot be considered a goal-directed instrumental task.

Thank you for this comment.

We originally described the task as a goal-directed instrumental task due to this description in similar discrimination tasks guided by sensory cues (Matteucci et al., 2022, *Neuron*, in press; Shiotani et al., 2020, *eLife*, 9:e57268; Courtiol & Wilson, 2016, *J.Neurosci*, 36(22):5946-5960); however, we agree that it is difficult to exclude the potential influence of Pavlovian conditioned cues in these (as well as our own) tasks, and have such removed mentions of “goal-directed behavior” from the manuscript.

Concerning the control of responses in the VD-Inhibit (previously VD-Avoid) task, while Pavlovian inhibition may be occurring towards the CS-, we believe that it is unlikely that an appetitive Pavlovian association could be formed between the random cues (CS+) and the reward since 51 random visual stimuli were presented for the CS+. Additionally, VD-Inhibit and VD-Attend tasks use the same random visual stimuli, so if a Pavlovian association between the random visual stimuli (CS+) and the reward was formed during the VD-Inhibit task, then we would expect to observe biased responding towards these cues following the transition to the VD-Attend task (as is seen in reversal learning tasks). However, there was no difference in %Correct between animals that were naïve to VD-Attend and those that had experienced VD-Inhibit task once (see figure below). Importantly, the finding that optogenetic inhibition of D2-MSNs during the outcome period of error trials in the VD-Inhibit task negatively influences task performance in the subsequent trial suggests that responses are, at least partly, under goal-directed control and influenced by error signaling on a trial-by-trial basis rather than purely influenced by Pavlovian associations (unless error signaling by D2-MSNs is able to transiently modulate the salience of the Pavlovian inhibitor).

(b) This is not an avoidance task. This paradigm relies on never pairing the flag with reward. The overall design used is best characterised as A+ X-; B- X+ where A is the marble and B is the Flag. X is a CS on its own which relies on randomly presented stimuli. It is hard to tell what specific associative process drives the responses because of inaccuracies of the design. Seems like the effect observed could rely on conditioned inhibition to the flag since it is not paired with reward, but in any case inhibitors cause withdrawal, not avoidance. Excitation, however, is clearly driven by A+ vs. AX- X- in the marble condition and by B- BX- X+ in the flag case; where, in the former, A (the marble) is a Pavlovian executor and, in the latter, B (the flag) is a Pavlovian inhibitor. Thus, the responses measured in this paradigm entirely depend on Pavlovian discrimination, and have nothing to do with avoidance behaviour. Avoidance is tactically where the response itself is what inhibits an aversive state (fear/frustrative non-reward). Here, what reduces the aversive state signalled by the flag is not well defined and remains ambiguous. Most likely, the force driving the response is the alternative stimulus, which is, of course, associated with reward, and not the response itself. On a related note, the flag and marble stimuli are not counterbalanced across tasks. Why not? Perhaps the flag is not very salient and so is a poor stimulus. If so, that alone would explain the difference in performance observed.

Thank you for this comment.

The term “avoidance” was used chosen based on its use in a similar cognitive neuroscience study in humans (Frank et al., Science 2004); however, we agree that this term has a different meaning in the field of behavioral neuroscience. As such, we have replaced "avoidance" with the term "inhibition" and have renamed the “VD-Avoid” task to the “VD-Inhibit” task.

As discussed above, the formation of a Pavlovian association between the random visual stimuli and the reward alone cannot explain responding in the VD-Inhibit task, as no biased responding towards the random visual stimuli (X-) was observed when the VD-Inhibit task (X+ vs. A-) was followed by the VD-Attend task (A+ vs. X-). As the reviewer suggests, it is possible that CS- in the VD-Inhibit task has become a Pavlovian inhibitor, and that this is driving responding in the task.

Regarding potential differences in the saliency of the visual cues, a previous study has reported that mice show no initial bias in responding towards the same flag and marble cues on the first session of a standard visual discrimination task (Morton et al., 2006, Nat. Methods, 3:767). Thus, there appears to be no difference in the saliency of the two visual stimuli.

A second potential major issue in this paper is the fact that they use D2-Cre mice. The field has long moved away from using D2R to drive the targeting to D2-expressing MSNs due to the fact that a shorter isoform of the receptor (D2s as opposed to D2l) is also encoded by the same promoter but is strongly expressed in pre-synaptic dopamine terminals. This means, in this case, that Cre is present in dopamine terminals in the striatum, which can be very problematic, especially considering the rather promiscuous tropism that AAVs have when infecting membranes in the neuropil (where even retrograde transport is not

uncommon, e.g. Aschauer et al 2013). Indeed, most data in Figure 2 (which are not immediately consistent with the predictions from the literature that D1- and D2-MSNs oppose each other) may very well be explained by this particular artefact. For example, it is not immediately logical that overall % of correct trials drops in both cases (Figure 2D). The way I'd interpret this, D1-iDREADDs mice drop performance because D1 neurons are inhibited (as intended; makes sense as these neurons go straight to SNr and are considered the "effectors" of the striatum), whereas D2-iDREADDs drop performance because dopamine drops (through DREADDs-mediated terminal inhibition), which consequently under-stimulates D1-MSNs—ultimately resulting in the same effect. Based on this, one could even provide a good alternative explanation for the differences seen between genotypes in performance after correct or after error trials (Fig. 2E and F). There, both D1-iDREADDs and D2-iDREADDs show the same performance drop after correct trials (this could be attributed to D1-MSN inhibition in the former and dopamine dropping in the latter, as above). On the other hand, both manipulations differ in "After Error" performance: D1-iDREADDs remains invariable while D2-iDREADDs drops again. This latter effect could indeed be due to the postsynaptic D2-MSN inhibition intended in the experiment, but it is overshadowed by the fact that "After correct" performance also goes down. To be fair with the conclusions provided here on error-based learning, the D2-iDREADDs group should show unaltered "After Correct" performance and blunted "After Error" performance. An experiment using A2a-Cre mice instead could reconcile these results and support the claim and main conclusion of the paper.

Thank you for this suggestion.

As the reviewer points out, there have been identified to be problems with using D2-Cre mice to investigate D2-MSNs due to the expression of D2s receptors, and therefore Cre, in the presynaptic terminals of DA neurons in the striatum. Thus, to confirm whether the reduced VD-Inhibit task performance observed in D2-Cre mice following optogenetic suppression in the outcome period of error trials is actually due to the effect of reducing activity in D2-MSNs, we have now performed this same experiment using A2a-Cre mice. Reassuringly, the results in the A2a-Cre mice closely matched those we had previously observed in the D2-Cre mice. These control experiments have now been added to the Supplementary information (see Supplementary Fig. S10).

Additionally, due to the difficulty in interpreting the iDREADDs experiments because of expression in both NAc core and shell regions, as well as leakage to the dorsal striatum, we have decided that it is best to remove this experiment from the manuscript. In its place, we have performed additional optogenetic experiments in D1-Cre mice, that alongside the optogenetic experiments in D2-Cre and A2a-Cre mice provide greater detail of the functional contributions of D1- and D2-MSNs to task performance at different time points (see Fig. 5 and Supplementary Fig. S11).

Other Comments

1 – I do not quite see how the post-error slowing effect mentioned in lines 79-84 (Figure S1) is relevant here, especially since the effect is not even significant. I may help removing this part altogether.

Thank you for this comment.

We have now removed this sentence.

2 – It should be acknowledged that both core and shell of the NAc received DREADDs as per Figure S2. The way information in lines 96-99 reads, one understands that only the Core was manipulated.

Thank you for this comment.

Due to leakage of iDREADD expression to the NAc shell and, to a lesser extent, the dorsal striatum, we have now decided to remove iDREADD experiments from the manuscript. The new inclusion of D1-Cre optogenetic experiments, alongside data from D2-Cre and A2a-Cre mice, largely negates the necessity of these iDREADD experiments for investigating the functional roles of D1- and D2-MSNs by alternatively investigating the effect of neural activity suppression in a time-locked manner to specific events in the task (ITI, cue, outcome).

3 – The rationale for comparing "After correct" and "after error" responses could be better articulated.

Perhaps a bit more literature on error-based correction in the literature would have helped highlighting the importance of these analyses.

Thank you for the helpful comment.

We have now cited several papers that have indicated D2-MSNs to be important for behavior modification after reward omission (Yawata et al., 2012; Nonomura et al., 2018; Tai et al., 2012; Tsutsui-Kimura et al., 2017) as a rationale for comparing performance in trials following correct or error responses (Lines 202-205).

4 – Related to my main comment above, I don't think that the data presented in Figure S5 supports the claim that "D2-MSNs are implicated in avoidance-based goal-directed behaviour following response errors". Here performance does not even change in "After Error" trials (Fig S5E).

Thank you for this comment.

Although the previous Figure S5 has now been removed as we have removed the iDREADD data from the manuscript, we saw the same task-specific contribution of D2-MSN error signaling in our optogenetics experiment (Fig. 5I and Suppl Fig. S11H) (i.e. no change in performance in A2a-Cre mice in the VD-Attend task).

In the current behavioral paradigm, we suggest that selection is based on the difference in the relative value of the two stimuli. In the VD-Inhibit task, while the action value of the random cues cannot be updated due to their random nature, the value of the flag decreases after each error response. The relative decline in the value of the flag cue prompts a response toward the random stimuli. In contrast, in the VD-Attend task, while the activation of D2-MSNs signals a response error, the CS- is a random cue so value information cannot effectively be combined with information concerning the specific cue to which the error corresponds and therefore does not influence task performance (the task is instead likely guided by learning of the relative increase in action value of the correct cue). Therefore, we believe that a task-selective contribution of D2-MSNs to inhibition-based choice behavior is observed.

5 – In Figure 3, why did the authors run the clustering analysis in the entire dataset only? It is indeed interesting to see that a general analysis pulled a large percentage of D2 cells as Type II, but I think this is only the beginning. This analysis should perhaps be followed by a second clustering study where D1 and D2 neurons are analysed separately. It would be very helpful to see the behaviour of all D1 neurons against the behaviour of all D2 neurons side by side in correct and error conditions - the statistical comparison between these datasets would be valid and very informative, particularly for appreciating the heterogeneity of functions that supports one system and the other in correct vs error trials. Furthermore, I think that such separate clustering arrangements would support the analysis conducted in Figure 4 more directly, where the aim is to identify the evolution of activity behaviour in identified neurons of each system throughout learning. As is, all analyses in Figures 3 and 4 are based on a general clustering analysis run on both populations lumped together, which is not immediately logical based on the core hypothesis of this study. To start with, D1-MSN and D2-MSN responses are not even collected from the same transgenic mouse strain: combining these datasets as one could have statistical concerns.

Thank you for this important comment.

In response to this comment and a similar comment from Reviewer2, clustering has now been performed separately for D1- and D2-MSNs. The new data are largely consistent with those observed in the previous clustering (notably, a large population of D1-MSNs activated during the outcome period of correct responses and a large population of D2-MSNs activated during the outcome period of error responses in the VD-Inhibit task), although several other interesting subpopulations are identified to be activated at various time points of the VD-Inhibit and VD-Attend tasks. The new data are presented in Figs. 2 and 3.

6 – In figure 5G, a significant trial type x light interaction ensuring that the drop of correct responses seen upon inhibition are indeed due to the trial type considered would be necessary (only the simple effect is reported in the figure legend).

Thank you for this comment.

The statistics for this figure were as follows. $F_{\text{interaction}} = 4.880$, $p = 0.0582$. However, this experimental design does not assume that there will be selective differences in the trials after error responses. For example, our previously included DREADD experiment required an interaction between treatment and genotype in the performances of the hM4Di and mCherry groups to confirm that CNO itself has no effect on behavior. On the other hand, for optogenetic experiments, since After-Correct and After-Error are compared under different conditions, an interaction is not necessary in this case. Since the treatment effect is significant, we can move on to the post-hoc multiple comparison test, where significant differences were detected in the performance of trials after error responses.

1 – Start of line 30: outcome instead of outcomes

Thank you for this comment.
The text has been revised.

2 – What is the "shaping" phase indicated in Fig 2C? I can't find this in the methods.

Thank you for this comment.
“Shaping” refers to pretraining. The term has now been changed into “pretraining” to fit with descriptions in the methods section of the manuscript.

3 - In Figure 3, I believe the normalised Z-score colorimetric scale under panel F belongs to panel E?

Thank you for this comment.
In the new figure, the position of the colorimetric scale was moved so that it is clear which figure the scale belongs to.

4 – Line 201: “Nevertheless, we detected a significant number of cells that acquired discriminability between correct and error outcomes (Fig. S10) in both cell types.” - I don't seem to be able to find these data in Figure S10...?

Thank you for this comment.
The figure has been modified (shaded panels included) so that it is easy to see the cells that have acquired the discriminability between the correct and error. To clarify the meaning of the auROC auxiliary line, an auROC histogram was added to the figure to show the fraction of cells that showed statistically significant activity (see Supplementary Fig. S7A&B).

5 – Line 245: models instead of model

Thank you for this comment.
The text has been revised.

Reviewer #2 (Remarks to the Author):

In this interesting work, authors used a new visual discrimination task in mice, and assessed the role of NAc D1-/D2-MSNs in cue-guided goal-directed “avoidance”. By performing cell-type specific neuronal silencing and in-vivo imaging, authors propose that NAc D2-MSNs to selectively contribute to cue-guided avoidance behavior, with activation of NAc D2-MSNs following response error playing an important role in optimizing future goal-directed behavior. They further suggest that error-signaling by NAc D2-MSNs underlies the ability to use environmental cues to avoid inappropriate behavior.

The methodology is adequate, the paper is well written and experimental controls are provided. The manuscript is interesting but has some methodological and conceptual issues that need to be

solved/clarified and some experiments performed in order to support the claims. Please find below my main concerns regarding the current version of the manuscript. I truly hope that these comments and suggestions will be useful for the authors and improve the status of the manuscript.

Major comments

1. Is avoidance a correct term? The term “avoid” is usually used to define an adaptive behavior in response to an aversive situation, including active and passive avoidance. In this task, the animal has to inhibit a particular behavior in order to get a reward, not to prevent something aversive, so I think it is not an adequate term to describe the task.

Thank you for this comment.

The term “avoidance” was used chosen based on its use in a similar cognitive neuroscience study in humans (Frank et al., Science 2004); however, we agree that this term has a different meaning in the field of behavioral neuroscience. As such, we have replaced "avoidance" with the term "inhibition" and have renamed the “VD-Avoid” task to the “VD-Inhibit” task.

2. Was imaging also performed in the attend task? This would be an interesting comparison to perform and would strengthen authors message.

Thank you for this comment.

Data for D1-/D2-MSN VD-Attend imaging has been added to the supplementary information (see Supplementary Fig. S5&6). This new data demonstrates the task-dependent contribution of NAc D1/D2-MSNs.

3. I have several concerns about the neuronal clustering methodology, but also the interpretation of the calcium imaging data (see below)

4. If D1 and D2 neurons have different relevance for correct vs error trial codification and to cue-guided “avoidance” behavior, then their activity should predict the outcome of a particular trial. authors assume this is the case because the majority of type II neurons are D2, but this is not proven directly with the analysis that is provided.

Thank you for your important comment.

We also are interested in whether the activity of D1- and D2-MSNs during the outcome period is able to accurately predict performance in the next trial; however, we were unable to perform prediction analysis of this data due to the small number of error trials at the late stage of learning.

5. Authors should also provide imaging data for D1- and D2- neurons alone (without merging the two populations), and perform clustering if the sample size is sufficient. This would allow to visualize D1 and D2 activity during task performance and to understand how many subpopulations exist of each neuronal subtype.

Thank you for this important comment.

In response to this comment and a similar comment from Reviewer1, clustering was performed separately on D1 and D2 and the results are shown in Figs. 2 and 3.

6. Viral expression in Fig. 5a shows significant spreading to the dorsal striatum, which can limit some of the interpretations of the paper. In addition, for DREADD experiments, a similar amount of virus was also used, and according to sup fig. 2, the transfection did not occur only in NAc core as authors report, but also shell and dorsal st. Since CNO was injected i.p., the behavioral results may derive from inhibition of other brain regions as well.

Thank you for this comment.

Due to leakage of iDREADD expression to the NAc shell and, to a lesser extent, the dorsal striatum, we have now decided to remove iDREADD experiments from the manuscript. The new inclusion of D1-Cre optogenetic experiments, alongside data from D2-Cre and A2a-Cre mice, largely negates the necessity of these iDREADD experiments for investigating the functional roles of D1- and D2-MSNs by alternatively investigating the effect of neural activity suppression in a time-locked manner to specific events in the task (ITI, cue, outcome).

7. Regarding optogenetic experiments, a key experiment that is missing is the manipulation of D1-MSNs. In addition, activation of D2 (and D1) neurons during the same periods could provide additional information about the role of each population in behavior. And what is the impact of manipulating these populations in different stages of the vd-attend task? I strongly believe that these experiments would strengthen the message that the authors want to convey with this manuscript.

Thank you for important comment.

As described above, we have now performed conducted optogenetic suppression experiments in D1-Cre mice. Suppression of D1-MSNs during ITI, cue, or outcome periods of the VD-Inhibit and VD-Attend tasks was not found to alter performance after correct or error trials (see Fig. 5D-F and Supplementary Fig. S11).

8. How are these results reminiscent of RPE-like?

Thank you for this comment.

The relevance of our findings to RPE is discussed within the discussion section (see Lines 311-330).

Methodology

9. Task design – since this manuscript is the first time that the task is performed, authors should provide detailed information:

- pretraining phase: a stimulus was randomly displayed – what type of stimulus? One of the 52 images described after? what is time out – light on like in the basic training session?; 77% of correct trials is based on what rationale?;
- How many days comprised each phase – provide distribution of performance so that new users now what to expect. (this is given in 1e but not for VD-Attend for example)

Thank you for this comment.

We have now added additional information about the pretraining sessions as well as referring the reader to the established visual discrimination task on which the design of these pretraining sessions was taken from (Horner et al., 2013, Nat Protocols, 8(10):1961-84). (see Lines 401-402)

Briefly, 51 images (the same as in the task training stage) were randomly presented in pretraining, and during the time-out the house lamp was switched on (as it was in the training sessions). The criterion of >75% of correct trials is recommended in a methods paper describing the same pretraining protocol used before a visual cue-based discrimination task (Horner et al., 2013, Nat Protocols, 8(10):1961-84). The number of days it takes for mice to pass each stage of pretraining was as follows: Must initiate=1 day, Must touch=1 day, Punish incorrect= 2-4 days.

We have now added this information to the Pretraining section of the materials and methods (see Lines 413-415).

Additionally, the performances of wild-type in the VD-Attend task have now been added to Supplementary Fig. S1.

- Why did you decided to have the lights on during the time out period? Since the light is turned on together with the tone, it will also be a CS.

Thank you for this comment.

We use house lamps without the tone to help the animal quickly recognize that it made an error response. Therefore, it is not a big problem for the task if the house lamp becomes a CS. This choice was made based upon the recommendation of a protocol paper for a touchscreen-based visual discrimination task (Horner et al., 2013,

Nat Protocols, 8(10):1961-84). As no significant change in neural activity was observed immediately after the start or end of the timeout, when the house lamp was illuminated, we believe the onset of the house lamp did not significantly modulate neural activity in D1- or D2-MSNs.

10. Z-score calculation: standard z-score, i.e., normalized to the signal of whole session or normalized to baseline of each trial? or other period?

Thank you for this comment.

Z-scores were normalized using values from -10 to +10 sec from the start of the trial. To better clarify the method, the above sentence was added to the materials and methods (see Line 447).

11. 1.2ul of viral delivery for GCaMP experiments is a very high amount. Why did you use this quantity?

Thank you for this comment.

The GCaMP virus was injected at two separate depths using a 4x dilution (600 nl per depth). We have now clarified this procedure in the materials and methods section (see Lines 381-384).

12. Cell registration methodology should be detailed.

Thank you for this comment.

The following sentence was added to the materials and methods (see Lines 457-462).
“To track the same cell in the early and late stages of learning, we compared the maximum intensity projection map of Session 2 (early) with that of the criterion session (late) within the same animals (Yang et al., 2022; Remedios et al., 2017) and registered identical cells using the plug-in function of Inscopix Data Processing software (NCC Score greater than 0.5).”

13. Fig S10: auROC methodology should be detailed.

Thank you for this comment.

The following sentence was added to the materials and methods (see Lines 453-456).
“The area under the ROC curve (auROC) was calculated from Z-scored neuronal activities on all correct and error trials every 100 ms for each of the cells. MATLAB's perfcurve function was used to calculate auROC.”

14. Statistics: did authors check for outliers?

Thank you for this comment.

We checked the normality of the data with Anderson-Darling test and performed parametric tests only on data sets for which a normal distribution was found. For other data sets, we applied nonparametric tests. The above information was added to the materials and methods (see Lines 476-477).

Results

15. Results of attend task should also be provided.

Thank you for this comment.

We have now added the data of wild-type performing the VD-Attend task (see Supplementary Fig. S1), D1- and D2-Cre mice in calcium imaging experiments during the VD-Attend (see Supplementary Fig. S5&6), and D1-, D2-, and A2a-Cre mice in optogenetic manipulation experiments during the VD-Attend (see Supplementary Fig. S11).

16. Lines 119-121: to say that manipulation of MSNs has an effect in motivation, authors would have to perform other type of experiments.

Thank you for this comment.

Given the leak of expression to the NAc shell and dorsal striatum, all iDREADD data and discussion was removed from the manuscript.

17. Regarding S4, there is increased latency to reward due to D1-MSN inhibition. Did authors controlled for locomotion due to this manipulation?

Thank you for this comment.

As above, due to the leak of expression to the NAc shell and dorsal striatum, all iDREADD data and discussion was removed from the manuscript.

18. S4: how can you have huge differences in earned rewards in D1 but not D2 animals since both present reduced % of correct due to the manipulation? This is because animals do not go and get the reward? It was not clear to me. If this is the case, then that reward becomes available for the next trial in the lick? To what session is S4 referring to?

Thank you for this comment.

As above, this figure has now been removed from the manuscript due to leak of iDREADDs to the NAc shell and dorsal striatum.

19. Are the animals of the attend session others or the same as the avoid task? This is not explicit.

Thank you for this comment.

The animals in the VD-Attend are the same as in the VD-Inhibit (previously VD-Avoid) task. To clarify, the following sentence was added to the materials and methods (see Lines 440-441).

“VD-Inhibit and VD-Attend tasks were performed in the same individuals, with VD-Inhibit always performed first.”

20. S5 e: I do understand that due to variability there are no differences in after error, but the variability is quite remarkable in comparison to previous graphs, showing that likely the manipulation has some effect as well.

Thank you for an interesting comment.

We assume that even in the VD-Attend task, to some extent, animals are using error information to modify their behavior. However, since it is difficult to draw conclusions from negative results, we do not discuss this in the text.

21. Not clear how the clustering was performed in fig 3. The description in methods is not clear. Please revise. 3 PCs refer to what?

Thank you for this comment.

To better clarify the method, the following sentence and a reference were added to the materials and methods (see Lines 448-453).

“Neuronal activity patterns were classified based on their activity patterns through a previously described unsupervised clustering approach (Cohen et al., 2012, Nature, 482:85-88). Briefly, the first three principal components of the Z-scored neuronal activities of all neurons averaged across all correct and error trials were calculated using principal component analysis (PCA), with the singular value decomposition algorithm. Hierarchical clustering of the first three principal components was then performed using a Euclidean distance metric and a complete agglomeration method. MATLAB's linkage function was used to perform the hierarchical clustering.”

22. If you perform the hierarchical clustering based on D1 neurons separately from D2 neurons, do you still observe the same clustering pattern?

Thank you for this comment.

We have now performed clustering separately for D1- and D2-MSNs. Although the number and percentage of clusters that could be identified changed slightly, the conclusion that D2-MSNs were more responsive to the outcome period of error trials remained the same.

23. Fig4D: which neuronal traces of this figure are representative of the posterior clusters?

Thank you for this comment.

The neural activity traces for before and after learning (now called “early” and “late”) have been revised to make them easier to understand (see Fig. 4).

24. Fig4e: you recorded 3 D1- and 3 D2- animals, how many cells per each animal?

Thank you for this comment.

The number of cells recorded from each animal is now listed in the Results section (see Lines 105-108).

25. Fig4F: absolute numbers should also be presented and not only %

Thank you for this comment.

The actual number of cells has now been added to the pie chart (see Fig 2F & 3B).

26. Fig4G-J it is not easy to identify ITI, cue periods – time stamps in x axis does not match what is said in the legend

Thank you for this comment.

The graphs have been redesigned and labels have been included so that each period can be easily identified (see Fig 2 & 3).

27. Fig4G-J – can you also provide the same type of graphs but separating D1- and D2-neurons?

Thank you for this comment.

We have now created new figures for the separated clustering of D1- and D2-MSNs (see Fig 2 & 3).

28. How did authors consider the time out periods in error trials in the imaging analysis?

Thank you for this comment.

Time Out is defined as the 5 seconds immediately following the error response. As described above (Response to Comment 9), no significant change in neural activity was observed immediately after the onset or offset of the timeout. Therefore, we concluded that D2-MSNs were activated by error but not by the house light itself.

29. Fig4G-J – Authors should also report in the results section (not only in figure legends) that types III and IV also present changes in the ITI period. These findings mean that even in non-relevant periods of the task, the activity of these neurons is already different between correct vs incorrect trials. How to explain this? Can this different activity predict subsequent trial outcome, i.e, correct vs incorrect?

Thank you for an interesting comment.

We have now added a description of each neuron Type’s activity during each time period to the Results section (see Lines 119-120 and 125-128. It is unclear why a change in the ITI is observed in some neuron types, although

it appears that this change is driven by increased or decreased activity during the ITI in error trials. Due to the smaller number of error trials compared to correct trials (as mice had reached criterion levels of responding where approximately 80% of trials were correct), it may be that there is more random variance in error trials than correct trials even when data is averaged across all trials of the same type. Indeed, our optogenetic experiments indicate that inhibition of this ITI period does not significantly alter the performance of the task.

30. Lines 168-170 – inhibition of neurons during a period does not necessarily mean these neurons are not required for the observed outcome, and this is particularly evident regarding GABAergic neurons as nucleus accumbens MSNs. Inhibition of MSNs can in turn disinhibit downstream target regions and promote a specific response. Thus, the assumption that D1 inhibition during error and activation during correct trials is mostly related to “correct-related signaling” is a biased interpretation of the findings.

Thank you for an important comment.

We agree that inhibition of D1-MSNs may be contributing to error signaling and have now included a discussion of this possibility in the discussion section (see Lines 262-271).

31. Fig 5B-E- signal iti, cue, outcome labels in all graphs

Thank you for this comment.

Labels for the ITI, Cue, and Outcome have now been added to all graphs. (see Fig 2 & 3, and S5 & S6).

32. S10: the auROC was calculated for the whole trial? Not clear

Thank you for this comment.

The average value between 2.0 to 3.5 seconds from the start of the Outcome period, corresponding to the peak of neural activity, was used. The above information was added to the legend as well as the methods section (see Lines 453-456 & 628-630).

33. Line 201-202 not sure what how you can infer from the data presented in S10 that cells acquire discriminability?

Thank you for this comment.

As learning progresses, D1 and D2-MSN in the NAc respond strongly to either correct or error trials. D1-MSNs have an equal occurrence of cells with a preference for correct and cells with a preference for errors, whereas D2-MSNs have an increased population of cells with a preference for errors. These cells acquiring discriminability between early and late sessions have now been marked by blue and red shaded areas in Suppl Fig S7.

Introduction/Discussion

34. Other relevant studies provided evidence about the role of D1- and D2-MSNs in reward and aversion-related behaviors and should be properly referenced in the adequate sentences (lines 44-47; lines 267 etc): doi: 10.1016/j.neuron.2015.08.019, doi: 10.1038/s41380-019-0484-3, doi:10.1016/j.celrep.2020.02.095, doi.org/10.1371/journal.pone.0207694, DOI: 10.1038/ncomms11829, amongst others.

Thank you for this comment.

The above papers have now been discussed and cited in the Discussion section (see Lines 299-308).

35. Earlier ephys studies regarding the role of NAc neurons (lines 296-294) should be mentioned such as those from Carelli’s lab. Though they do not segregate between D1 and D2 neurons, these earlier studies show very robustly the role of NAc neurons in this type of processes

Thank you for this comment.

The relevance of several papers from the Carelli lab to our own data have now been discussed within the discussion section (see Lines 289-298).

Minor comments

36. line 410 typo

Thank you for this comment.

I looked for a typo but could not find one. Can you point out more specifics?

37. figures could be improved. For example fig 2 CNO individual dots should be colored according to the condition – grey red and not in the same color as it is now – other option is to include Vehicle CNO in the x axis.

Thank you for this comment.

The DREADD data has been removed from the manuscript. However, we have made an effort to try to make the optogenetics figures easier to understand for the reader by adding colors to the “laser on” sections of each figure (see Fig 5, S10, and S11).

38. Scales of graphs should be consistent for the same type of result

Thank you for this comment.

We modified all graphs so that the scales are consistent.

39. Fig 2d bottom refers to which session?

Thank you for this comment.

We have now removed this DREADD figure from the manuscript.

40. Number of animals in sup fig 4?

Thank you for this comment.

We have now removed the DREADD data from the manuscript.

41. lines 187-194: authors could improve the description of these results, I suggest to be more systematic in the way they present information.

Thank you for this comment.

The description in this section was revised (see Lines 157-178).

We hope this revision provides a better explanation of our data.

42. Videos – which animals? D1/D2?

Thank you for this comment.

Both videos are D2-Cre mouse data. We have now added this information to the legend (see Lines 704-708).

43. Mentioning expert mice and novice mice gives the impression that the data are not actually from the same animal across learning phases. Authors could use early vs late learning?

Thank you for this comment.

Following the reviewer's suggestion, the text was changed to “Early” and “Late”. (see Lines 157-183 and 627-632).

REVIEWER COMMENTS

Reviewer #1 (Remarks to the Author):

The authors have done a good job at addressing my concerns. While discussions about what specific associative phenomena actually support this task could go on forever, I am satisfied with the new terminology and the amendments made (e.g. removing the "goal-directed behavior" term and changing task name to "VD-Inhibit"). I also value the addition of a new experiment on A2a-Cre mice to end the problem of unrestricted expression of the transgene to pre-synaptic dopamine terminals. In fact, the authors now find an effect on "After Error" performance only, which better supports the hypothesis. All my other comments have also been addressed, I believe this manuscript is now ready to be published.

Reviewer #2 (Remarks to the Author):

Authors improved the manuscript by performing additional optogenetic manipulation of D1-MSNs, added VD-attend data and separated clusters of D1- and D2-MSN subpopulations. Authors also decided to remove DREADDs experiment from the manuscript. However, there are still important issues that need to be clarified.

1. The number of animals per group and total number of cells is low, compared with other studies of the same brain region (e.g. van Zessen, 2021 eLife). Considering the variability of responses, this can be a major drawback for authors' conclusions. The reduced n of animals is even more problematic because at least 2/6 animals have the lenses implanted outside the core (S2).
2. Methodological information is missing, authors should explain in detail all the procedures, also some figures are not properly described (eg. S8).
3. Optical activation experiments of D1- and D2-MSNs would strengthen the conclusions of the manuscript.
4. There are neuronal clusters of D2 neurons that also present changes in activity during ITI period (and D1 as well). These findings show that the activity of these neurons is already different between correct vs incorrect trials throughout the task, and not only in outcome-specific periods. How can you conciliate this with your hypothesis?
5. PCA and clustering data should be provided, tSNE would be very informative (even if in Sup. Material). What was the criteria used to determine the number of clusters? How good was the clustering? Agglomerative coefficient? Silhouette scores?
6. Clustering analysis: was the entire period of the trial included in the analysis or were just the isolated periods of ITI, Cue and Outcome analyzed? If the latter is true, it would be relevant to include the whole trial period.
7. A statistical comparison between the D1- and D2 clusters datasets in correct and error conditions would be important to perform.
8. Provide cue duration of the VD inhibit task – not clear?

9. Why did authors consider the cue period as 0-1.5 sec from trial initiation and -1.5-0sec from a response? It is not clear.
10. A table with all statistical analysis and values should be provided.
11. Did authors checked if tracked neurons in Fig. 4B-J reflect the activity of all neurons? I.e., If authors take the activity from tracked neurons of 4B for example (D1 type I) of criterion phase, is this activity statistically not different from the activity of all neurons in 2G in the criterion phase? And the same applies to all clusters. Also, what is the number of cells tracked in each type?
12. In data from figure 4 it would be more relevant to see differences in the PSTH (check at which moment of the task are significant differences) rather than compare average AUC (as is represented in Figure S7) of in Figure 4K,L,M,N. Also, it would be interesting if the cells formed new clusters or moved from one cluster to other.
13. Lines 172 – 173, 181-182 are statements not supported by the available data in my opinion
14. Why is auROC calculated in a different period than the one used before for outcome period?
15. How did you calculate discriminability? On what ground did you select the low and upper limits?
16. S8: authors should indicate the right nomenclature for each color eg. D1-Arch1 and not “optx”
17. Regarding the opto suppression experiments, for me it is not clear the experimental design. Animals were subjected to 3 consecutive sessions, each with stim as described in S9. If so, how can you compare between first and last trials? Because animals get satiated, tired etc... maybe this is the reason why you have animals with performances so low, and this is a major caveat in the interpretation of the findings, in my opinion.
18. In the opto experiments, D1 animal presents less than 50% of correct responses in the VD-Inhibit task, which is not expected. There is so much variability in D1 animals that removing one animal would probably give a difference in cue inhibition. Did authors check for outliers throughout experiments?
19. How many error trials exist in each condition of Fig 5? This is important to know as in 5F for example and 5I (and S10E), in the error condition, there are animals with 100% of correct responses, whereas this is not evident in the after correct condition in any of the other graphs.
20. Because authors only stimulated 50% of the trials, did authors controlled for the effect of stimulation in the performance of subsequent trials? I think that just by separating between “after correct” and “after error” is not enough, as there might be bias in the effects depending on the number of trials with stimulation.
21. In the VD-attend opto experiments (S11), the variability in performance is dramatically different (higher) than in VD-inhibit task, so it is difficult to state that manipulation of D1 or D2 neurons does not affect task performance. There are several animals with low performance, so how can we really be sure that the opto manipulation is not affecting performance? Just as an example, animals are required to have >80% correct in 2 days or >75% in 3 days before opto manipulation, and there are animals with 20% correct responses in the D1 inhibition S11C for example – this is a dramatic decrease. Can authors comment on this? If authors plot these same graphs (of these same animals) in the days before opto manipulation, what would you have in after correct and after error?
22. Line 499 – 5 clusters
23. Response latencies should be provided. Number of rewards earned should be provided.

Reviewer #1 (Remarks to the Author):

The authors have done a good job at addressing my concerns. While discussions about what specific associative phenomena actually support this task could go on forever, I am satisfied with the new terminology and the amendments made (e.g. removing the "goal-directed behavior" term and changing task name to "VD-Inhibit"). I also value the addition of a new experiment on A2a-Cre mice to end the problem of unrestricted expression of the transgene to pre-synaptic dopamine terminals. In fact, the authors now find an effect on "After Error" performance only, which better supports the hypothesis. All my other comments have also been addressed, I believe this manuscript is now ready to be published.

We thank the reviewer for their careful check of the manuscript and are glad that they are satisfied with the changes.

Reviewer #2 (Remarks to the Author):

Authors improved the manuscript by performing additional optogenetic manipulation of D1-MSNs, added VD-attend data and separated clusters of D1- and D2-MSN subpopulations. Authors also decided to remove DREADDs experiment from the manuscript. However, there are still important issues that need to be clarified.

1. The number of animals per group and total number of cells is low, compared with other studies of the same brain region (e.g. van Zessen, 2021 eLife). Considering the variability of responses, this can be a major drawback for authors' conclusions. The reduced n of animals is even more problematic because at least 2/6 animals have the lenses implanted outside the core (S2).

We thank the reviewer for their comment.

While the number of cells in our manuscript is lower than that in the paper cited by the reviewers, the cited paper uses a very high number of cells in comparison to many previously published studies using miniscope-based imaging of specific cell populations in deep brain regions (n = 56-79 cells from 4-5 mice in VP_{GABA}, n = 80-88 cells from 3 mice in VP_{Penk}, n = 37-48 cells from 2-3 mice in VP_{Glu}, Heinsbroek et al., 2020, Cell Reports; n = 361 cells from 3 mice in saline, n = 356 cells from 3 mice in cocaine, Zhao et al., 2022, Science Advances). We believe that our data (n = 259 cells from 3 mice in D1-Cre, n = 194 cells from 3 mice in D2-Cre) is within the normal range used for imaging experiments in deep brain regions in mice.

Concerning the placement of the lenses. While the tip of two of the GRIN lenses were located at the boundary between the Core and the lateral shell, the actual focal plane of the lenses is approximately 300 μm below the tip of the lens (as indicated in Fig 2B), which sits within the NAc core. However, in the interest of caution, we have now changed sentences describing the "NAc core" to the "dorsolateral NAc"

2. Methodological information is missing, authors should explain in detail all the procedures, also some figures are not properly described (eg. S8).

Thank you for this comment.

Corrections have been made to the Methods and the terminology in Fig.S2 and S8 (current Fig. S13). If further corrections are needed, could you please point out the specifics?

3. Optical activation experiments of D1- and D2-MSNs would strengthen the conclusions of the manuscript.

Thank you for this comment.

While the experiment that the reviewer proposes is potentially interesting, it is unlikely that such manipulations would result in any observable improvements in performance given that mice are already responding at criterion levels at the time that optogenetic manipulations would be performed (post-learning). We believe that, in combination with the imaging data, the inhibitory optogenetic experiments currently included in the manuscript sufficiently demonstrate the involvement of D2-MSNs in signaling responses errors necessary for successful task performance.

4. There are neuronal clusters of D2 neurons that also present changes in activity during ITI period (and D1 as well). These findings show that the activity of these neurons is already different between correct vs incorrect trials throughout the task, and not only in outcome-specific periods. How can you conciliate this with your hypothesis?

Thank you for this interesting comment.

It has been reported that some neurons in the NAc encode the average or net expected reward in a given block (Wang et al. Nat Neurosci. 2013; Ito and Doya. J Neurosci. 2015). Therefore, we analyzed whether the results of the previous trial contributed to the neural activity of the current trial. We found that the D1-Type I and Type IV neuronal activity, which showed differences in neural activity in the ITI, reflected the results of the previous trial (Fig. S6). On the other hand, D2-MSN activities were found to be determined by the combination of the results of the previous trial and the current trial (Fig. S7). Considering that optogenetic suppression of D1-MSNs during ITI period affected the performance, it is possible that the NAc activity during ITI also contributed to the update of the action values in the next trials. A discussion of these results has been added to the manuscript (L120-125, L133-144, and L292-304).

5. PCA and clustering data should be provided, tSNE would be very informative (even if in Sup. Material). What was the criteria used to determine the number of clusters? How good was the clustering? Agglomerative coefficient? Silhouette scores?

Thank you for this comment.

A PCA and tSNE plot demonstrating the validity of the hierarchical clustering were created and added to the Fig. S5. The clustering used in the current study is based on an established method for clustering of neural activity and reported in Cohen et al. (Nature 2012) and Stephenson-Jones et al. (Neuron 2020). In this method, an arbitrary number of clusters is generally used, but this time, based on the tSNE values, the `evacluster` function in MATLAB was used to calculate the optimal solution among the numbers from 1 to 5, resulting in 5 clusters in D1- and 4 in D2-MSNs, which are consistent with the number of our clusters. We also calculated the silhouette score based on the euclidean distance (see figure below), and although a part of them had negative values because they were calculated using different criteria than hierarchical clustering, the vast majority (88%) of the total neurons had positive values, indicating that even based on a different index, the majority of neurons were correctly clustered.

6. Clustering analysis: was the entire period of the trial included in the analysis or were just the isolated periods of ITI, Cue and Outcome analyzed? If the latter is true, it would be relevant to include the whole trial period.

Thank you for this comment.

This clustering was done using the values of the whole trial.

7. A statistical comparison between the D1- and D2 clusters datasets in correct and error conditions would be important to perform.

Thank you for this comment.

D1- and D2-MSNs activities were clustered separately, as pointed out in the previous revision. This made it difficult to compare D1/D2 activities between the same clusters. To address this comment, we made a brute force comparison of D1/D2 activities in each time window (see figure below). However, we found it difficult to interpret anything from this data and did not add it to this paper. If you think a comparison between specific types would be useful, could you point out the specifics?

8. Provide cue duration of the VD inhibit task – not clear?

Thank you for this comment.

Visual cues were presented until mice responded at either window. We have confirmed that the above sentence is already in the Methods (L464).

9. Why did authors consider the cue period as 0-1.5 sec from trial initiation and -1.5-0sec from a response? It is not clear.

Thank you for this comment.

Cue duration varied from trial-to-trial (as the cue was present until the mouse responded) and little change was observed in between the 0-1.5 sec onset and -1.5-0 sec offset. Therefore, in order to standardise the analysis of this cue period between trials and mice, we analysed only these onset and offset periods together.

10. A table with all statistical analysis and values should be provided.

Thank you for this comment.

A table with all data values and statistics was added (Source Data).

11. Did authors checked if tracked neurons in Fig. 4B-J reflect the activity of all neurons? I.e., If authors take the activity from tracked neurons of 4B for example (D1 type I) of criterion phase, is this activity statistically not different from the activity of all neurons in 2G in the criterion phase? And the same applies to all clusters. Also, what is the number of cells tracked in each type?

Thank you for this comment.

For all clusters, we have now performed a comparison of the traces of the original and cell-registered clusters. The results confirmed that all clusters except D1-Type III (which was removed from subsequent analysis) were not statistically different from the original cluster (L177-179 and Fig. S10). In addition, the number of cells has been added to the new Fig. 4 and 5.

12. In data from figure 4 it would be more relevant to see differences in the PSTH (check at which moment of the task are significant differences) rather than compare average AUC (as is represented in Figure S7) of in Figure 4K,L,M,N. Also, it would be interesting if the cells formed new clusters or moved from one cluster to other.

Thank you for this comment.

In response to this comment, comparisons of neural activity in ITI and Cue periods before and after the learning were made for all clusters (L180-212 and Fig. 4 and 5). In addition, an analysis of the cluster transition was performed and added to the supplementary Figures (Fig. S11 and S12).

13. Lines 172 – 173, 181-182 are statements not supported by the available data in my opinion

Thank you for this comment.

We have now removed these statements from the manuscript.

14. Why is auROC calculated in a different period than the one used before for outcome period?

Thank you for this comment.

We have now removed all auROC analyses and replaced them with z-score analyses to make analysis methods consistent throughout the manuscript.

15. How did you calculate discriminability? On what ground did you select the low and upper limits?
Thank you for this comment.

As mentioned above, all auROC analysis and statements about discriminability have now been removed.

16. S8: authors should indicate the right nomenclature for each color eg. D1-Arch1 and not "optx"
Thank you for this comment.

The figure has been revised.

17. Regarding the opto suppression experiments, for me it is not clear the experimental design. Animals were subjected to 3 consecutive sessions, each with stim as described in S9. If so, how can you compare between first and last trials? Because animals get satiated, tired etc... maybe this is the reason why you have animals with performances so low, and this is a major caveat in the interpretation of the findings, in my opinion.

Thank you for this comment.

Only one session (of a particular stimulation type (ITI, cue, or outcome)) was performed per day, and sessions were performed on consecutive days in a pseudo-randomized order (latin-square). We have now clarified this better within the text and in the methods (L228-233 & L482-487).

18. In the opto experiments, D1 animal presents less than 50% of correct responses in the VD-Inhibit task, which is not expected. There is so much variability in D1 animals that removing one animal would probably give a difference in cue inhibition. Did authors check for outliers throughout experiments?

Thank you for this comment.

In response to this comment, mice with values greater than 2 SD from the mean were removed from the analysis as outliers. As a result, we were able to detect a significant effect of optogenetic suppression of D1-MSNs on the performance in the VD-Inhibit, as Reviewer 2 commented. Based on this result, the interpretation of the results and the Discussion regarding the function of D1-MSNs was revised (L242-244; L292-304).

19. How many error trials exist in each condition of Fig 5? This is important to know as in 5F for example and 5I (and S10E), in the error condition, there are animals with 100% of correct responses, whereas this is not evident in the after correct condition in any of the other graphs.

Thank you for this comment.

A figure showing the number of trials in each condition was added (Fig. S20).

20. Because authors only stimulated 50% of the trials, did authors controlled for the effect of stimulation in the performance of subsequent trials? I think that just by separating between "after correct" and "after error" is not enough, as there might be bias in the effects depending on the number of trials with stimulation.

Thank you for this comment.

In response to this comment, we analyzed the impact of ITI and Cue suppression on the performance of the next trial, but no impact on the next trial was observed. For outcome stimulation, the data presented in Fig. 6D-F is the response in the next trial. These findings have been added to Results (L246-248 & Fig. S17).

21. In the VD-attend opto experiments (S11), the variability in performance is dramatically different (higher) than in VD-inhibit task, so it is difficult to state that manipulation of D1 or D2 neurons does not affect task performance. There are several animals with low performance, so how can we really be sure that the opto manipulation is not affecting performance? Just as an example, animals are required to have >80% correct in 2 days or >75% in 3 days before opto manipulation, and there are animals with 20% correct responses in the D1 inhibition S11C for example – this is a dramatic decrease. Can authors comment on this? If authors plot these same graphs (of these same animals) in the days before opto manipulation, what would you have in after correct and after error?

Thank you for this comment.

We acknowledge that for an unknown reason the variability in the VD-Attend task is higher than in the VD-Inhibit task. However, a comparison between performance on the last day of training and in each of the

sessions with LED stimulation for VD-Attend task (see figure below) revealed no significant changes in performance. Thus, despite this variance we believe that it would have been possible to detect a significant change in performance as a result of optogenetic inhibition if one had occurred.

22. Line 499 – 5 clusters

Thank you for this comment.

The text has been revised.

23. Response latencies should be provided. Number of rewards earned should be provided.

Thank you for this comment.

Response latencies and number of earned rewards have been added to the Results (L248-251 and Fig. S18 and S19).

REVIEWERS' COMMENTS

Reviewer #2 (Remarks to the Author):

The authors have answered to my comments. I believe that the manuscript is greatly improved and the message is strengthen with the new data/analysis. I would like to congratulate the authors for the work.

Ana Joao Rodrigues